# Indolent primary cutaneous B-cell lymphomas resemble persistent antigen reactions without signs of dedifferentiation

Johannes Griss [1] ✉, Sabina Gansberger [1], Inigo Oyarzun [1], Martin Simon[1], Mathias C. Drach[1], Vy Nguyen[1], Lisa E. Shaw [1], Ulrike Mann[1], Stefanie Porkert[1], Matthias Farlik[1], Wolfgang Weninger[1], Werner Dolak [2], Bertram Aschenbrenner [1], Beate M. Lichtenberger [1], Shawn Ziegler-Santos[1], Christine Wagner[1], Ingrid Simonitsch-Klupp[3], Stephan N. Wagner [1], Constanze Jonak [1] & Patrick M. Brunner [4] ✉

Primary cutaneous B-cell lymphoma encompass clinically heterogeneous entities. While primary cutaneous diffuse large B-cell lymphoma, leg type (pcDLBCL-LT) is aggressive, primary cutaneous follicle centre lymphoma (pcFCL) and primary cutaneous marginal zone lymphoma (pcMZL) typically follow an indolent course. To clarify their pathophysiological basis, we perform single-cell RNA sequencing on pcFCL, pcMZL, and pcDLBCL-LT, alongside reactive B-cell rich lymphoid proliferations (rB-LP), gastric mucosa-associated lymphoid tissue (MALT) lymphoma, and systemic counterparts. Here we show that the indolent pcMZL, pcFCL, and rB-LP exhibit a persistent germinal centre reaction, not observed in pcDLBCL-LT or gastric MALT lymphoma. Further, pcMZL top expanded clones develop within lesions from naïve and not post-germinal centre B cells as currently presumed. Our data thus indicate that pcMZL and pcFCL, similar to rB-LP may be driven by (a yet unknown) antigen. While our data indicates that pcFCL exhibits some features of true lymphomas, it clearly supports the classification of pcMZL as a lymphoproliferative disease.

Primary cutaneous B-cell lymphomas comprise a heterogeneous group of extranodal non-Hodgkin lymphomas[1]. The most common subtypes include primary cutaneous marginal zone lymphoma (pcMZL), primary cutaneous follicle centre lymphoma (pcFCL), and primary cutaneous diffuse large B-cell lymphoma, leg type (pcDLBCL-LT)[1]. pcMZL and pcFCL are generally indolent conditions with a 5-year disease-specific survival of >95%[2]. In contrast, pcDLBCL-LT is an aggressive disease with 5-year survival rates between 20 and 60%[3].

For pcMZL, this clinical observation led to a reclassification to a lymphoproliferative disorder in the International Consensus Classification (ICC) joint classification particularly referring to their class-switched cases[2]. These comprise IgG or IgA positive B-cell proliferations, which follow an indolent clinical course. pcMZL may be preceded by reactive B-cell rich lymphoid proliferations (rB-LP, formerly called "pseudolymphoma"), matching that pcMZL are not veritable lymphomas[4]. In line, several studies showed that only around 5% of B cells within pcMZL lesions correspond to a top expanded clone[5,6]. Nevertheless, the WHO classification currently maintains the term lymphoma for pcMZL leading to a conflict between these two classifications[7].

[1]Department of Dermatology, Medical University of Vienna, Vienna, Austria. [2]Division of Gastroenterology and Hepatology, Department of Internal Medicine 3, Medical University of Vienna, Vienna, Austria. [3]Department of Pathology, Medical University of Vienna, Vienna, Austria. [4]Department of Dermatology, Icahn School of Medicine at Mount Sinai, New York, NY, USA. ✉e-mail: johannes.griss@meduniwien.ac.at; patrick.brunner@mountsinai.org

Previous gene expression analyses identified characteristic B-cell phenotypes for each primary cutaneous B-cell lymphoma subtype. pcMZL B cells were found to be most similar to plasma cells and pcFCL showed a more germinal centre (GC)-like phenotype, whereas pcDLBCL-LT lesions were most consistent with activated B cells[8–10]. In cutaneous B-cell lymphomas, clonality was not always detectable using routine molecular methods, or when a surrogate such as light-chain restriction was used[11]. In those cases, the differentiation from rB-LP can be difficult[12]. To distinguish pcMZL from rB-LP, skin flow cytometry was found to have superior sensitivity than immunohistochemistry, in situ hybridization or immunoglobulin heavy chain gene rearrangement examinations[12]. However, its routine use is difficult as it requires fresh tissue, is prone to technical challenges during processing, and is not widely available[12]. Apart from a more precise molecular characterisation of the disease, the terminology of lymphoma versus lymphoproliferative disorder or cutaneous lymphoid hyperplasia impacts patient education and especially the patients' perception of their disease[13,14].

Here, we present the largest single-cell RNA sequencing characterisation of primary cutaneous B-cell lymphomas and rB-LP and compare these findings to data from their non-cutaneous B-cell counterparts. In this work we show that the indolent pcMZL, pcFCL, and rB-LP exhibit a persistent germinal centre reaction, not observed in pcDLBCL-LT or gastric MALT lymphoma. This indicates that these entities may be driven by (a yet unknown) antigen. Moreover, our data clearly supports the classification of pcMZL as a lymphoid proliferation.

## Results

### scRNA-seq based characterisation matches the current histopathological understanding of primary cutaneous B-cell lymphomas

We obtained fresh biopsies from patients diagnosed with pcMZL (n = 9), pcFCL (n = 5), pcDLBCL-LT (n = 4), or cutaneous rB-LP (n = 5), and normal skin from healthy volunteers (NHS, n = 4) (Supplementary Table 1). Samples were analysed using single-cell RNA-sequencing (scRNA-seq) coupled with B-cell receptor (BCR) sequencing. After quality control, we obtained data for 268.224 cells (pcDLBCL-LT: 16.908, pcFCL: 71.720, pcMZL: 109.028, rB-LP: 51.336, NHS: 19.232).

Using canonical markers (Supplementary Fig. 1a), we were able to identify T cells and NK cells, B cells, plasma cells, blood vessel and lymphatic endothelial cells, dendritic cells, plasmacytoid dendritic cells, macrophages, melanocytes, keratinocytes, smooth muscle cells, and fibroblasts (Fig. 1a, Supplementary Fig. 1b).

We further subclustered all B cells in order to arrive at a detailed phenotypic characterisation (Fig. 1b). This revealed naïve B cells, as identified by *CD19*, *MS4A1* (CD20), and *IGHD*, germinal centre (GC) B cells, as identified by *CD38*, *AICDA*, and *BCL6* that were further subdivided into a light zone (LZ) and dark zone (DZ) GC B-cell population, based on the presence or absence of *MKI67* (Fig. 1c). Memory B cells were characterised by their expression of *CD19*, *MS4A1* (CD20), and *CD27* with the absence of *CD38* and *BCL6* (Fig. 1c). Plasma cells were identified through the expression of *CD27*, *CD38*, *SDC1* (CD138), and the lack of *MS4A1* (CD20) (Fig. 1c). Thus, we were able to identify the full spectrum of canonical B-cell phenotypes ranging from naïve B cells to plasma cells.

In addition, we observed aberrant B-cell populations, all originating from pcDLBCL-LT, that formed distinct clusters and could not be classified using canonical B-cell markers (Fig. 1b–e). These were in part *MS4A1* (CD20)+, but also expressed *MKI67* without the typical expression of *CD38* and *CD27* for canonical germinal centre B cells ("aberrant B1", Fig. 1c). Additionally, we observed *MS4A1* (CD20) negative B cells that lacked the expression of *SDC1* (CD138) and high levels of immunoglobulin associated genes that would be expected for plasma cells ("aberrant B2", Fig. 1c). This indicates the pcDLBCL-LT contains B cells that no longer match the physiological B-cell phenotypes.

We subsequently performed a differential expression analysis of all of these subtypes (Supplementary Data 1). Matching the phenotypic assignment, naive B cells primarily showed an up-regulation of activation and signalling associated genes, such as *CD69*, *CD83*, and *CD44*, as well as *CXCR4* which is important for chemotaxis and tissue homing. Moreover, they expressed *FCER2* (CD21) linked to B cell maturation[15]. Next to immunoglobulin associated genes, genes highly expressed in plasma cells were mainly associated with protein synthesis, such as *DNAJB9*, *DNAJC3*, *PDIA4*, and secretion, such as *SEC61A1*, *SEC24D*, and *SAR1*. Germinal centre cells showed a strong expression of cell cycle regulating genes, such as *PLK1*, *CDC20*, *CCNA2*, *CCNB1*, *CCNB2*, *CDK1*, and *CDCA3*. Finally, the group of aberrant B cells uniquely overexpressed energy metabolism associated genes, such as member of the electron transport chain for ATP production (*NDUFA4*, *NDUFB11*, *NDUFB4*, *NDUFB7*, *NDUFB10*, *NDUFB9*, *NDUFV2*, *NDUFA11*, *NDUFS6*, *NDUFS5*), but also oxidative phosphorylation (*COX5A*, *COX7B*, and *COX6A1*) and glycolysis (*GAPDH*, *LDHA*, *LDHB*, *TPI1*, *ENO1*, *PGAM1*). This clearly shows that these aberrant B cells were metabolically highly active, matching their proposed malignant phenotype.

To further validate our findings we re-processed publicly available scRNA-seq data from a study on the effect of oncolytic viral therapies on cutaneous B cell lymphomas by Ramelyte et al.[16] which contained one pcFCL and one pcDLCBL-LT sample (Fig. 1h, Supplementary Fig. 2). Matching our own data, the pcFCL sample's clone only consisted of germinal centre B cells (Fig. 1h) while clonally expanded B cells from the pcDLBCL-LT samples showed aberrant phenotypes (Supplementary Fig. 2). These cells were largely *CD27* positive with large portions expressing *MKI67*. Yet, the expected expression of *CD38* was lacking. Similarly, naive or plasma cells were only found in the polyclonal B cell fraction. Therefore, this data fully matches our own observations.

### pcMZL originates from pre-germinal centre naïve B cells and follows the canonical B-cell differentiation trajectory

Through our BCR sequencing data, we were able to clearly attribute B cells to the top expanded clone in each disease (Fig. 1d). In each sample, we were thus able to unambiguously distinguish the clonally expanded (presumed malignant) B cells from the polyclonal bystander infiltrate (Supplementary Data 2).

Matching current classifications[17], pcMZL and rB-LP uniquely contained relevant numbers of plasma cells (Fig. 1e). Conversely, pcFCL samples showed a dominance of LZ GC B cells (Fig. 1e). pcDLBCL-LT uniquely contained aberrant B-cell phenotypes and lacked relevant numbers of naïve B and canonical plasma cells (Fig. 1e). Our data is therefore in-line with the current histopathological understanding with distinct phenotypic compositions of the B-cell infiltrate for each entity.

pcMZL top clones are considered to derive from post GC B cells[17,18]. However, we found that clonally expanded B cells from our pcMZL samples spanned all canonical peripheral B-cell phenotypes, from naïve B cells up to terminally differentiated plasma cells (Fig. 1e). We used monocle3 to perform a pseudotime analysis of this data, which matched the canonical B cell differentiation (Fig. 1f, Supplementary Fig. 3). An analysis of the clonally expanded B cells of each pcMZL sample highlighted that the clone seemed to differentiate in each individual sample (Fig. 1g). This was consistent regardless of whether the sample exhibited a predominance of plasma cells or germinal centre B cells and was not limited to specific subtypes of pcMZL as previously described[18]. This suggests that expanded pcMZL clones develop from naïve, pre-GC B cells irrespective of the phenotypic composition.

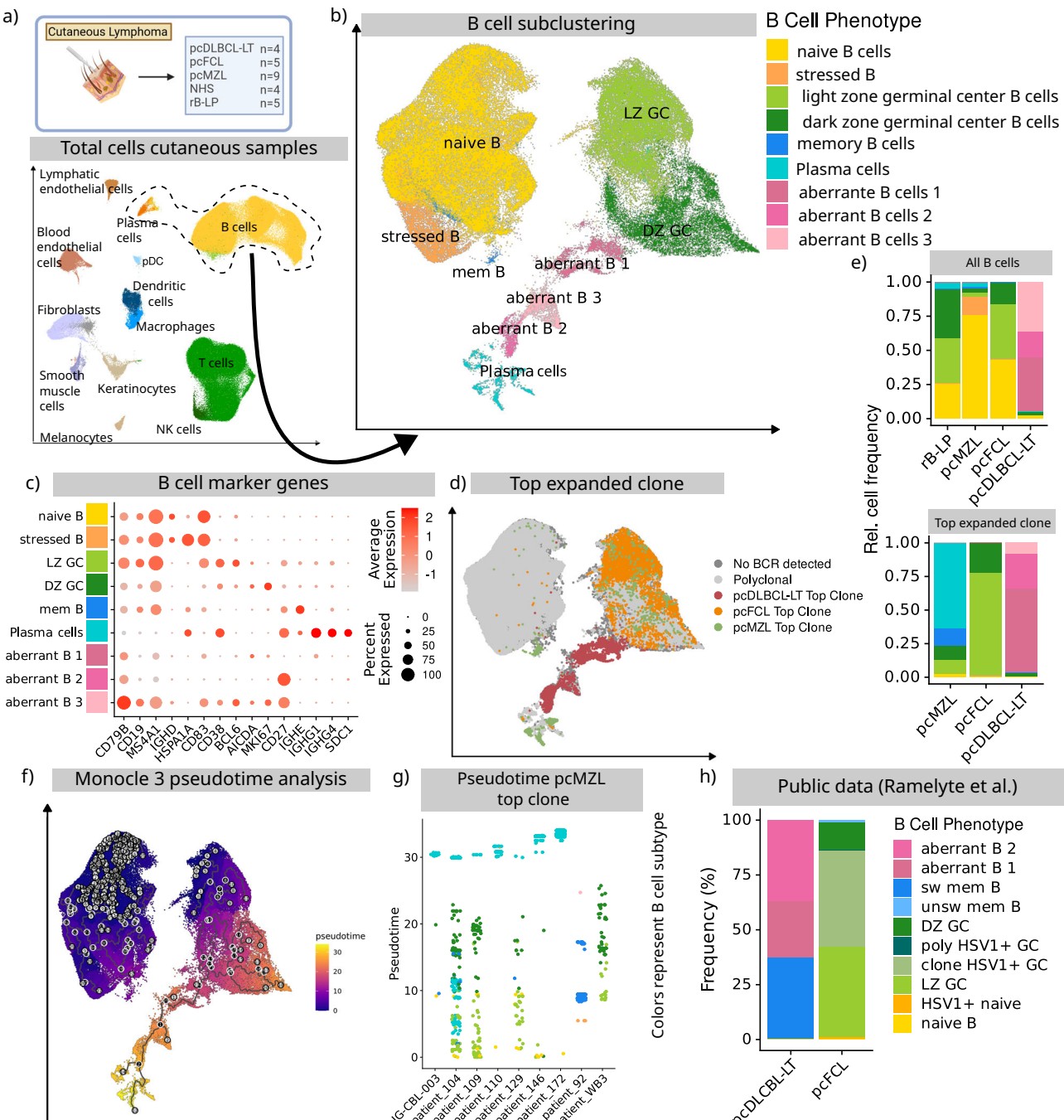

**Fig. 1 | scRNA-seq based characterisation of cutaneous B cells. a** Schematic overview of all cutaneous samples and UMAP embedding of the respective scRNA-seq data. Created in BioRender. Griss, J. (2026) https://BioRender.com/33vew3u. **b** UMAP embedding of the subclustered B cells (including B, and Plasma cells) of all cutaneous samples. **c** Dot plot of key B cell markers used to identify respective B-cell subtypes. Colour intensity represents average expression and point size the fraction of expressing cells. **d** UMAP embedding of the subclustered B cells highlighting the results of the BCR sequencing. Colours represent the top-expanded clone of all samples per disease. **e** Frequency of B-cell subtypes per disease shown for all B cells (top) and only B cells part of the top-expanded clone (bottom). **f** UMAP embedding of the B cell subclustering highlighting the results of the monocle3-based pseudotime analysis. Black lines represent the identified trajectories. **g** Pseudotime of each individual cell from the top-expanded clone in pcMZL per sample. Colours represent the identified B-cell subtypes. **h** Frequency of B-cell subtypes of the top expanded clone from one patient with pcFCL and one patient with pcDLCBL-LT from the study of Ramelyte et al. Source data are provided as a Source Data file.

## Multiplex immunohistochemistry-based characterisation validates scRNA-seq derived composition of B cell infiltrates

To corroborate our scRNA-seq findings, we used a multiplex immunohistochemistry (IHC) approach to simultaneously characterize naïve-like B cells (CD19+/CD20+, CD27−, CD38−, CD138−, CD5−), GC-like B cells (CD19+/CD20+, CD27−, CD38+, CD138−, CD5−), memory-like B cells (CD19+/CD20+, CD27+, CD38−, CD138−, CD5−), and plasma cells (CD19+, C20−, CD138+, CD5−) in a total of 40 samples (Fig. 2a).

Matching our scRNA-seq data, pcMZL and rB-LP samples harboured a diverse lymphocytic infiltrate with the highest number of plasma cells (Wilcoxon rank-sum test, FDR corrected $p < 0.01$), when compared to pcFCL and pcDLBCL-LT (Fig. 2b, c). By contrast, pcFCL

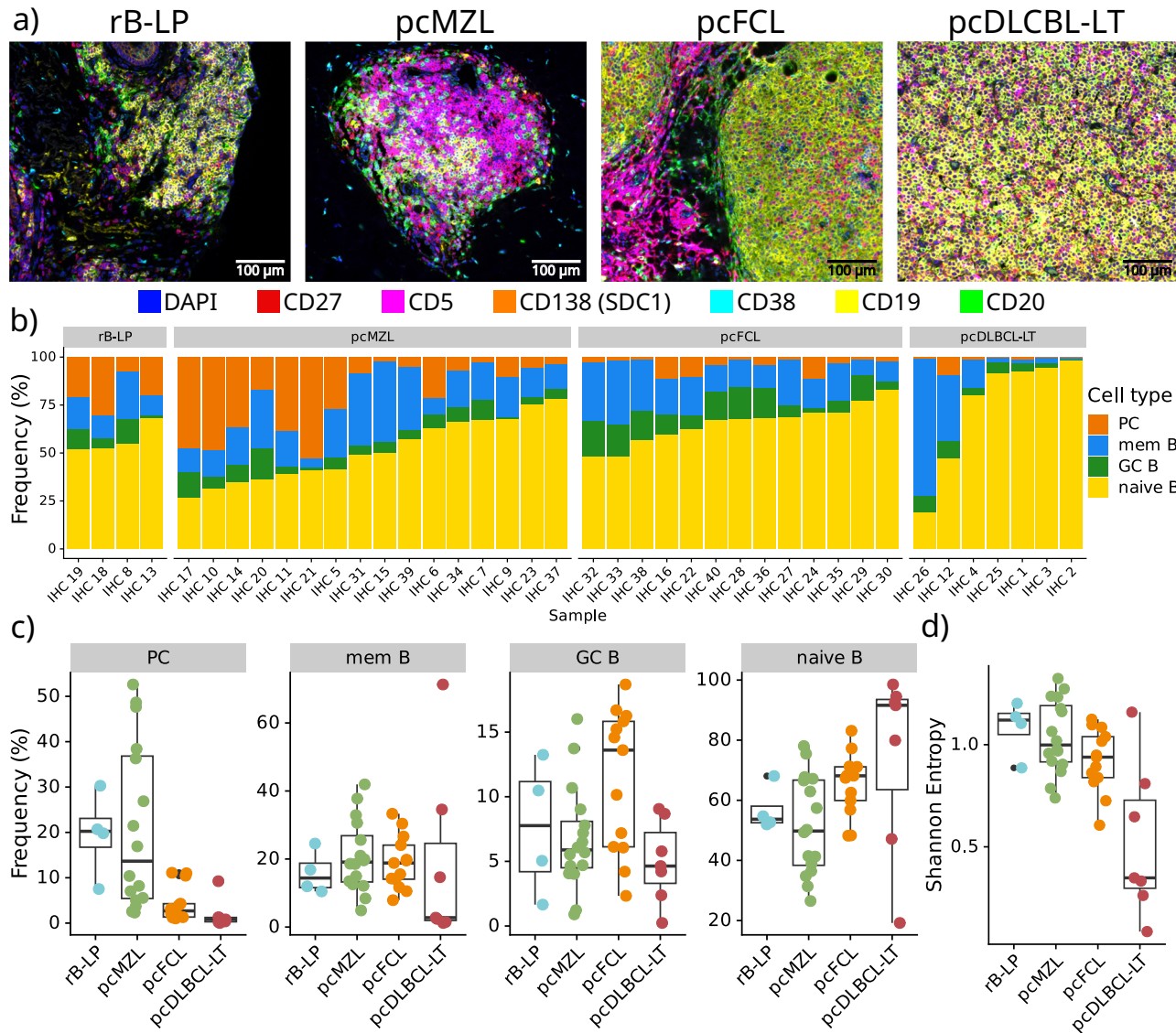

**Fig. 2 | Multiplex IHC-based characterisation of the B cell infiltrates in primary cutaneous B cell lymphoma samples. a** Representative combined images for all investigated entities from a total of 40 samples. **b, c** Relative abundance of the characterised cell types in all samples (*n* = 40). **d** Shannon entropy for each sample (*n* = 40) per disease. Lower and upper hinges of the boxplots correspond to the first and third quartiles, centre represents the median, whiskers extend to a maximum of 1.5 of the interquartile range and represent the minimum and maximum value within this range.

samples showed significantly more germinal centre B cells than all other entities (Wilcoxon rank-sum test, FDR corrected *p* < 0.01). While most pcDLBCL-LT samples were dominated by naïve B cells, some additionally contained high numbers of memory B cells. This directly confirmed our scRNA-seq data as IHC 26 was acquired from the same patient as the scRNA-seq samples patient_117 and patient_207 where we also observed high numbers of CD27+ (aberrant) B cells, similar to the samples from Ramelyte et al. [16] (Supplementary Fig. 2). A Shannon diversity index based analysis subsequently confirmed a significantly larger heterogeneity of lymphocyte composition in rB-LP, pcMZL, and pcFCL samples compared to pcDLBCL-LT (Wilcoxon rank-sum test, FDR corrected *p* < 0.01, D). Overall, these data corroborate our scRNA-seq based characterisation and shows the observed distributions seem stable across larger sample numbers.

### Primary cutaneous lymphomas show distinct cellular compositions from their systemic counterparts

It is yet unclear whether our observed phenotypic composition is typical of all B cell lymphomas or specific to CBCL. We thus acquired biopsies from 4 patients diagnosed with gastric mucosa associated lymphoid tissue (MALT) lymphomas refractory to *H. pylori* eradication (Supplementary Table 1), since this is one of the most frequent marginal zone lymphomas[19], and has long been considered the gastric correlate of cutaneous MZL[20]. Additionally, we integrated public data from studies characterising 18 samples from patients diagnosed with sFCL[21], 3 samples of reactive lymph nodes (RLT)[21] and 4 samples of sDLBCL[22] (Fig. 3a). This resulted in an integrated dataset of a total of 207,634 B cells. Based on canonical markers and differentially expressed genes, we identified all B cells observed in our cutaneous samples, as well as *ITGAX*+, age associated B cells (ABC)[23] (Fig. 3b, Supplementary Data 3).

The expanded clone in gastric MALT lymphoma samples (MALT) was uniquely dominated by *CD27*+ memory B cells. These showed a class-switch to either IgG or IgA (Fig. 3e), and low numbers of plasma cells (Fig. 3d). B cells of MALT distinctively expressed the genes *CD1C* and *CD1D* (Fig. 3f, Supplementary Data 4), which are relevant to lipid antigen processing[24]. Overall, MALT exhibited a significantly different phenotypic composition compared to pcMZL (Fig. 3g).

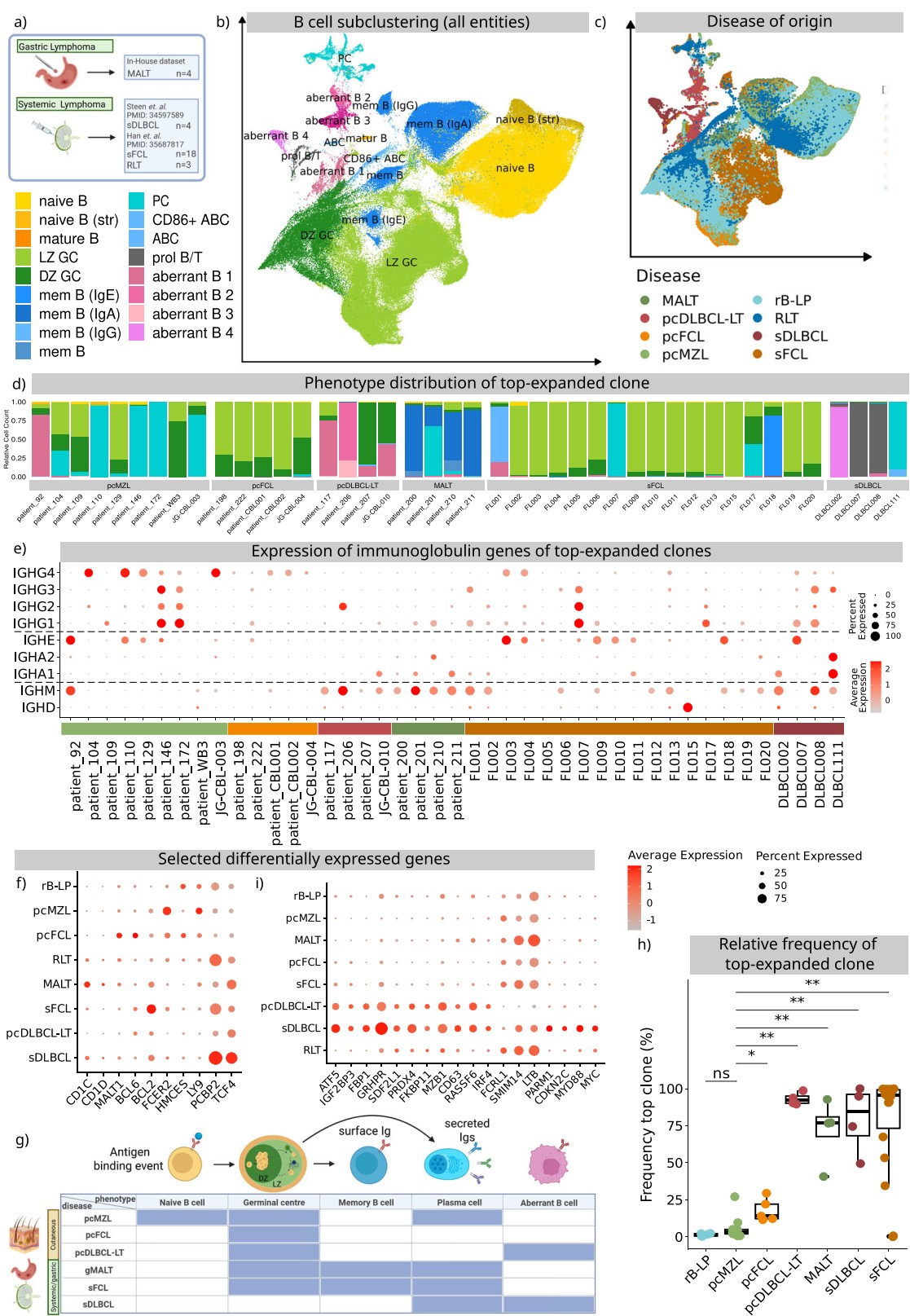

pcFCL was the only entity where more than 95% of the clonally expanded B cells were limited to germinal centre B cells (Fig. 3d). In sFCL, the heterogeneity between samples was considerably larger with several sFCL samples also containing clonally expanded memory or plasma cells (Fig. 3d). Matching current literature and diagnostic criteria, pcFCL B cells expressed high levels of *BCL6* and sFCL B cells *BCL2* (Fig. 3f). Moreover, pcFCL showed the highest expression

of *MALT1*, a proposed target for multiple B cell lymphomas. Otherwise, pcFCL exhibited overlaps with pcMZL in terms of differentially expressed genes (Supplementary Data 4). For example, B cells from both entities expressed *FCER2* (CD23), linked to B cell activation and antigen presentation[25], *HMCES*, which limits the accumulation of deletions during somatic hypermutation[26], and Lymphocyte Antigen 9 (*LY9*), a member of the SLAM family of immunomodulatory

**Fig. 3 | Integration of B cells from CBCL samples with B cells from MALT, sFCL, sDLBCL, and reactive lymph nodes. a** Overview of processed samples. Created in BioRender. Griss, J. (2026) https://BioRender.com/8i86vna. **b** UMAP embedding of all processed B cells (cutaneous and systemic lymphomas). **c** Disease of origin of the processed B cells. **d** Distribution of B cell phenotypes of topclone per sample grouped by disease. **e** Dot plot showing the abundance of immunoglobulin isotype genes in the top expanded clone per sample. **f** Dot plot showing expression values of genes differentially expressed between indolent cutaneous entities (rB-LP, pcMZL, pcFCL) and the other entities. **g** Visual representation of the canonical B cell trajectory and the respective distribution of the top expanded clones per disease. Created in BioRender. Griss, J. (2026) https://BioRender.com/8p4kb6z. **h** Relative proportion of the top expanded clone per sample ($n = 49$) and disease. Values are normalised based on the total number of B cells. Stars represent significant levels based on a two-sided Wilcoxon rank sum test with FDR correction (*<0.01, **<0.05, n.s. >0.05). Adjusted *p*-values for the comparison against pcMZL were 0.06 (rB-LP), 0.014 (pcFCL), 0.004 (pcDLBCL-LT, MALT, sDLBCL), and 0.003 (sFCL). Lower and upper hinges of the boxplots correspond to the first and third quartiles, centre represents the median, whiskers extend to a maximum of 1.5 of the interquartile range and represent the minimum and maximum value within this range. **i** Dot plot showing expression values of genes differentially expressed between pcDLBCL-LT and sDLBCL and other entities. Source data are provided as a Source Data file.

receptors which is associated with activation and differentiation in T cells[27] (Fig. 3f, Supplementary Data 4). Genes associated with tumour progression, such as *PCBP2* and *TCF4*, linked to high risk sDLBCL were significantly down-regulated in indolent CBCL[28] (Fig. 3f). Overall, indolent CBCL expressed genes associated with inflammation and immune cell differentiation, while lacking classical tumour promoters.

## pcMZL shows significantly lower rates of clonal expansion than other cutaneous and systemic lymphomas

Clonal expansion is considered a hallmark of lymphomas and can be tracked in B cells through BCR sequencing[29]. We therefore used the BCR data to compare the rate of clonal expansion between cutaneous and systemic lymphomas (Fig. 3h). In pcMZL and rB-LP, clonally expanded B cells represented a median of 4% and 2%, respectively, of all B cells, matching previous reports[5]. Outliers observed were likely due to sequencing artefacts (Supplementary Fig. 4). This was in contrast to pcFCL and pcDLCBL-LT which showed median clonal expansions of 52% and 97%, respectively. Similarly, in gastric MALT lymphoma the expanded clone accounted for a median of 77% of the total B-cell infiltrate. Moreover, sFCL and sDLBCL were similarly dominated by a single expanded clone, with a median expansion of 96% and 85% respectively. To test the heterogeneity between body sites we acquired two samples simultaneously from a patient with rB-LP (Supplementary Fig. 5). This analysis showed that the estimated clonal expansion was comparable with 6% and 9% between both sites (Supplementary Fig. 5). Additionally, we performed separate analysis of consecutively acquired samples within 6 and 12 months from a patient with rB-LP and pcDLBCL-LT respectively (Supplementary Fig. 6). In both cases, consecutive samples showed matching results with respect to phenotypic composition and rates of clonal expansion (Supplementary Fig. 6). This underscores both the robustness of our method and the temporal consistency of the diseases. Overall, the low rate of clonal expansion in pcMZL substantiates the theory that this condition is more similar to a lymphoproliferative disorder than a true lymphoma.

## DLBCL is characterised by aberrant B cell phenotypes that are not found in other entities

Clonally expanded B cells in pcDLBCL-LT could not be annotated following the canonical B-cell development in our samples (Figs. 1c and 3b, c). To arrive at a more detailed characterisation, we performed a subclustering of only pcDLBCL-LT B cells (Supplementary Fig. 7). We observed a loss of *MS4A1* (CD20) without the expression of *SDC1* (CD138) or CD38 as expected of CD20- plasma cells (Supplementary Fig. 7). There were two clusters of proliferating (*MKI67*+) B cells. While one matched DZ GC B cells, the other was *MS4A1* (CD20)-, with no expression of CD38 or CD27 required to define a germinal centre B cell (Fig. 3b, d, aberrant B2). These findings were consistent with the publicly available scRNA-seq data from Ramelyte et al.[16] (Supplementary Fig. 2). This highlights that the presence of aberrant B cell phenotypes is typical of pcDLCBL-LT.

Similar to pcDLCBL-LT, sDLBCL also showed clonally expanded aberrant B cells. This matched the results of a differential expression analysis where sDLBCL and pcDLBCL-LT B cells showed similarities compared to all other entities (Supplementary Data 4). These included the transcription factors and potential oncogenes *ATF5* and *IRF4*, the tumour suppressor *RASSF6*[30], the known DLBCL marker *IGF2BP3*[31], metabolism and protein synthesis associated genes such as *FBP1*, *GRHPR*, *SDF2L1*, *FKBP11*, and *MZB1*, genes protecting against oxidative stress such as *PRDX4*, and genes associated with survival signals such as *CD63* (Fig. 3i). At the same time, *FCRL1* which regulates BCR signalling was uniquely lost in pcDLCBL-LT and sDLBCL B cells (Fig. 3i). Simultaneously, the analysis also revealed differences between sDLBCL and pcDLBCL-LT matching their separate clusters. sDLBCL B cells uniquely expressed the potential leukaemia oncogene *PARM1*[32], the known ABC-type DLBCL marker *MYD88*, the B cell proliferation associated marker *MYC*[33], and the tumour suppressor *CDKN2C*[34] (Fig. 3i). pcDLCBL-LT B cells exclusively lacked the poorly characterised gene *SMIM14*, but also Lymphotoxin Beta (*LTB*), which is crucial for the development of lymphoid tissues[35]. This highlights that pcDLBCL-LT is distinct from the indolent CBCL and more similar to its systemic counterpart.

## A subset of non-class switched pcMZL samples also showed aberrant phenotypes, as opposed to class-switched pcMZL

Several authors currently differentiate between a class switched and non-class switched variant of pcMZL, where only the non-class switched variant is considered a true lymphoma[2,17]. In our series, one sample (patient 92) had a clear IgM+, but, surprisingly, also an IgE+ phenotype among the clonally expanded B cells (Fig. 3e). As only IgM and IgG are routinely assessed, this sample would therefore be classified as non-class switched pcMZL in conventional pathological assessments[2]. Surprisingly, this population contained aberrant B cells (Fig. 3d). Similar to B cells found in pcDLBCL-LT, these cells were *MS4A1* (CD20)−, *CD27*+, *IGHM*+, with a *MKI67*+ subpart that showed low expression of *CD38* and *AICDA* but no relevant expression of *IGHG1-4-* (Supplementary Fig. 8). There were no B cells in this sample that expressed CD138 (*SDC1*), which would be expected for CD20- B cells (Supplementary Fig. 8). While many aspects of these cells, such as proliferation and potential somatic hypermutation (expression of *AICDA*) is reminiscent of germinal centre B cells, the lack of CD20 is incompatible with this phenotype, matching reports of CD20 loss in systemic B cell lymphoma[36]. Clinically, patient 92 has shown an indolent course with two cutaneous recurrences and no sign of systemic disease within a total follow up of 12 years. Therefore, our data suggests that this sample's clone does primarily consist of aberrant, non-canonical B cells yet with no apparent influence on this patient's clinical course.

A second sample, WB3, also showed a non-class switched clone (Fig. 3e). Nevertheless, these clonally expanded cells matched canonical germinal centre B cells (Supplementary Fig. 8). In this sample, plasma cells were all polyclonal and IgG4 positive (Supplementary Fig. 8). In a routine histopathological assessment, sample WB3 would

therefore be classified as a class-switched pcMZL, as plasma cells would likely have been deemed the pathogenic cell population. This observation highlights that in this specific sample, the histopathological differentiation of class switched *vs.* non-class switched pcMZL is likely not yet precise enough to distinguish clinically relevant subtypes.

## Indolent B-cell lymphomas and rB-LP show similar interactions resembling inflammatory reactions

We performed an interaction analysis using CellChat to further characterise the mechanistic background of CBCL subtypes (Fig. 4a, Supplementary Data 5). All entities showed an interaction between *CXCL12* on BEC and FB, APP on most cells of the tumour microenvironment (TME), and *THBS1* on BEC which recruit B cells via CXCR4, CD74[37], and CD47 respectively[38]. *MIF*, a further key chemoattractant via CXCR4 in inflammatory reactions[37], was uniquely missing in pcDLBCL-LT. Only pcMZL B cells interacted with PODXL and CD34 on BEC, FB, and LEC via SELL, which allows cell migration[39]. All B cells expressed *IL16* which recruits other immune cells via CD4. Certain interactions were only found in indolent CBCL. For example, B cells interacted with each other through *PTPRC* and *CD22*, which regulates the threshold for lymphocyte activation and thereby controls inflammatory reactions[40]. Similarly, the interaction between *FCER2A* (CD23) which is key to B cell differentiation[41] and *ITGAX* on DC and MAC was only present on indolent subtypes. Finally, pcDLBCL-LT B cells uniquely interacted with *CD27* on B, T, and MEL through *CD70* which act as co-stimulatory molecules. Overall, this data matches the current hypothesis that pcDLBCL-LT B cells originate from antigen experienced B cells, while rB-LP, pcMZL, and pcFCL show similar interaction patterns that resemble inflammatory reactions.

## Indolent B-cell lymphomas and rB-LP are characterised by ongoing germinal centre reactions

The germinal centre reaction, which consists of a mechanism of ongoing somatic hypermutation, is a key part of B-cell function. We used IgBLAST[42] to quantify the rate of somatic hypermutation in all cutaneous samples (Fig. 4b). In all diseases, the top expanded clone showed a higher rate of somatic hypermutation compared to the polyclonal B cells. In pcMZL and rB-LP, the most expanded clone had the widest range of mutations. This matches the assumption that these B cells entered the germinal centre reaction and subsequently acquired mutations to improve antigen specificity, which is in line with the results of our pseudotime analysis of pcMZL samples (Fig. 1g). pcFCL similarly showed increasing rates of somatic hypermutation, yet starting at a considerably higher level (Fig. 4b). This matches our observation that the clone in pcFCL did not contain any naïve B cells (Fig. 1e). Yet, there is evidence of an ongoing acquisition of mutations. Clonally expanded B cells in pcDLCBL-LT showed the highest rates of somatic hypermutation but spanning a much smaller range (Fig. 4b). Despite our characterization of germinal centre B cells in pcDLBCL-LT samples, these data suggest that these cells are largely no longer acquiring new mutations typically associated with the physiological germinal centre reaction. This finding further supports the aberrant phenotypes observed in these samples. In MALT lymphoma samples, polyclonal B cells showed a similar rate of somatic hypermutation as the polyclonal B cells in pcMZL and pcFCL. Yet, the top expanded clone centred around a single value of somatic hypermutations. This matches our phenotypic data where clonally expanded B cells in MALT samples primarily consisted of post-germinal memory B and plasma cells that no longer undergo the germinal centre reaction (Fig. 2e, f). This data suggests that the three indolent cutaneous B-cell reactions, rB-LP, pcMZL, and pcFCL uniquely retain an intact germinal centre reaction.

The germinal centre reaction depends on several other cell types, most importantly *CXCR5* + T follicular helper cells (Tfh) and follicular

dendritic cells (fDC)[43]. We therefore subclustered the T cells from all cutaneous samples (Fig. 4c). We were able to identify *CD8*+, *PRF1*+, *IFNG*+ Tc1 polarised cells (CD8 Tc1), some of which were *LAG3*+ as a sign of exhaustion (CD8 Tc1 exh, Fig. 4d). Further, we observed *CD4*+, *CCR7*+, *SELL*+ CD4 central memory T cells (Tcm), *FOXP3*+, *IL2RA*+ regulatory T cells (Treg), *MKI67*+ proliferating T cells (prol T), *PDCD1*+, *CTLA4*+ exhausted CD4+T cells (CD4 Th2 exh), as well as *CXCR5*+ T follicular helper cells (Tfh). Tfh were only present in rB-LP, pcMZL, and pcFCL samples (Fig. 4e), matching previous reports where Tfh were histologically observed in all cases of pcMZL[44].

Matching their mesenchymal origin with an assumed fibroblast-like progenitor[45], fDC are part of fibroblast clusters in scRNA-seq data. We therefore further subclustered all fibroblasts from the cutaneous samples and identified *APCDD1*+, *COL18A1*+ secretory papillary (spFB), *CCN5* +, *SLPI* +, *DPP4*+ secretory reticular (srFB), *PARD3B*+, *FBXL7*+, *AUTS2*+ presumably migratory (migFB), *COL11A1*+, *MMP11*+, *POSTN*+ mesenchymal (mesFB), *SFRP4*+ myo (myoFB), *APOE*+*CCL19*+ inflammatory (iFB) fibroblasts, *CR2*+, *FCER2*+, *CR1*+*CXCL13*+ follicular dendritic cells (fDC) and *S100B*+ *SOX10*+ Schwann cells (SC) (Fig. 4f, g). Similarly to Tfh, fDC were only observed in rB-LP, pcMZL, and pcFCL samples (Fig. 4h). Therefore, with the detection of an ongoing somatic hypermutation and the presence of Tfh and fDC, samples of rB-LP, pcMZL, and pcFCL uniquely contained all required aspects of a functional germinal centre reaction (Fig. 5).

## Discussion

The indolent behaviour of pcMZL, pcFCL, and rB-LP has sparked discussions on their actual nature, and led to conflicting classifications with respect to pcMZL[7]. Previous studies showed that patients' quality of life is directly influenced by whether a disease is classified as a lymphoma or not[14]. This highlights that we need to arrive at a clear statement concerning the nature of these entities for patients, as well as for clinical practice.

Our data presents consistent evidence that the investigated three indolent B-cell diseases, rB-LP, pcMZL, and pcFCL are uniquely characterised by ongoing germinal centre reactions. We were able to detect both continuous somatic hypermutation, as well as the required support cells in these samples. Extra nodal germinal centre reactions are generally considered a sign of the formation of tertiary lymphoid structures[46], are primarily observed in cancer[47] and autoimmune diseases[48] and are a sign of chronic, antigen-specific immune responses[49], especially in the skin[50]. This matches multiple reports where both pcMZL and pcFCL were linked to infections[51–53] and responded to antibiotic therapies or vaccinations[54–56]. Next to the ongoing germinal centre reaction, the top expanded clones in pcMZL represented only a low fraction of the overall B cells, matching previous reports[5]. This low number of pathogenic clones is in stark contrast to all other lymphomas and matches reports that (systemic) DLBCL is driven by antigen-independent B-cell receptor activation[57]. Additionally, clonally expanded pcMZL B cells followed physiologic B-cell trajectories, originating from naïve B cells, counter to current hypotheses and in contrast to MALT and pcDLBCL-LT[17]. This further aligns with previous studies that were unable to find consistent driver mutations in pcMZL[17] in contrast to pcDLCBL-LT[58]. Overall, pcMZL therefore does not show any of the characteristics found in the other investigated B-cell lymphomas but matches rB-LP with the sole difference that pcMZL developed a presumably antigen-directed, expanded clone. We therefore present molecular evidence that aligns with the concept that pcMZL is a lymphoproliferative disorder and not a true lymphoma.

Our data further questions the proposed class-switch based classification of pcMZL into a lymphoproliferative disorder or a lymphoma[4]. In our data, one non-class switched sample may contain aberrant B cells, yet this patient showed an indolent course within a

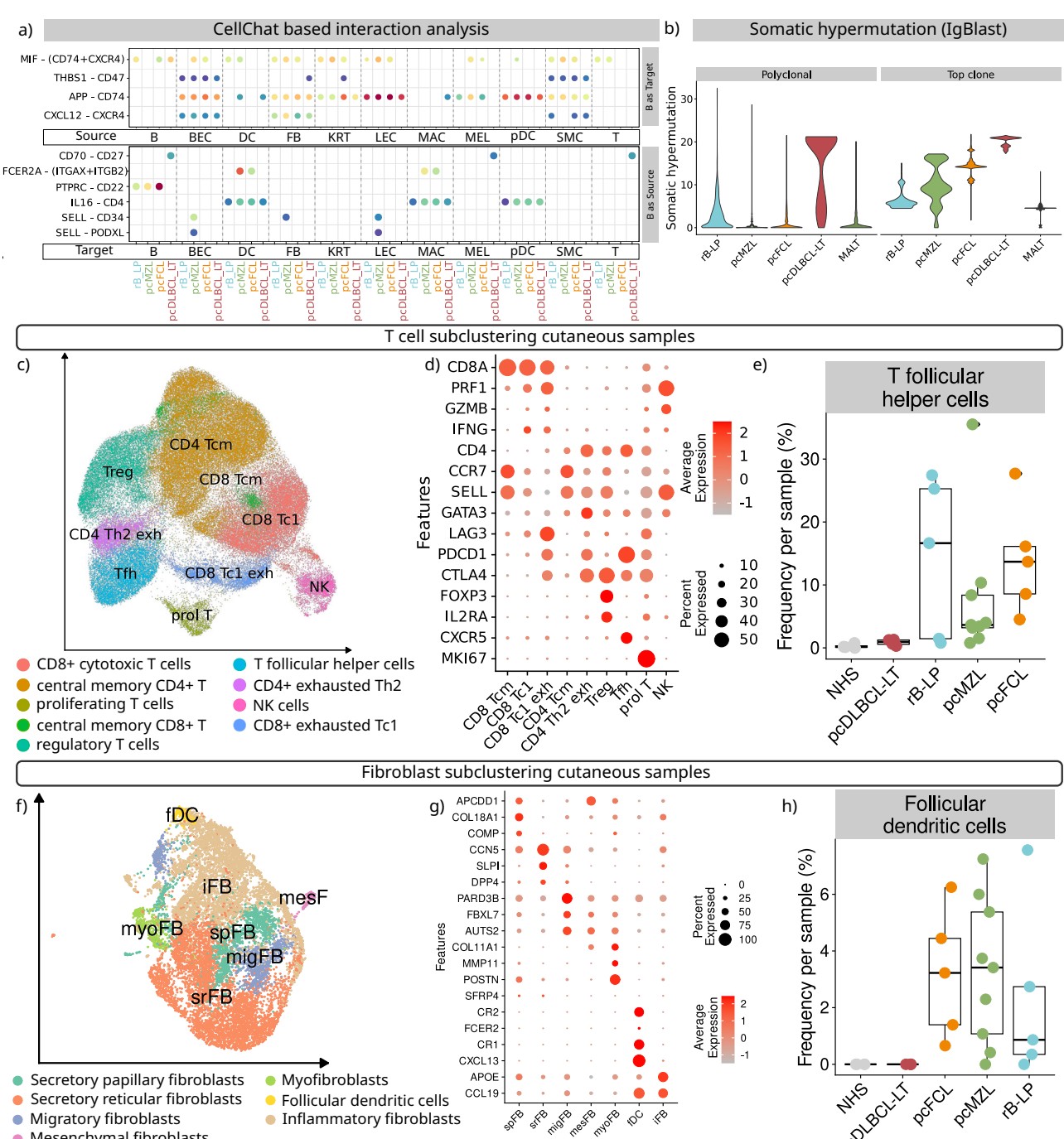

**Fig. 4 | Indolent primary cutaneous B cell lymphoma show an intact germinal centre reaction. a** Key cell-to-cell interactions based on CellChat. Colour coding represents the cumulative interaction probability and size the associated CellChat-based *p*-value. **b** Somatic hypermutation in the top expanded B-cell clone and polyclonal B cells per disease as derived through the identity mapping of IgBLAST. Somatic hypermutation is quantified through the difference to the canonical sequence. Data is shown separately for the top expanded clone and the polyclonal infiltrate. **c** UMAP embedding of a subclustering of the T cells from all cutaneous samples. **d** Canonical markers used to identify key T cell subtypes. Size of the dots represent the percentage of cells expressing the marker, colour intensity represents the average expression. **e** Relative proportion of T follicular helper cells

among all identified T cells per sample (*n* = 27) (coloured points). Lower and upper hinges of the boxplots correspond to the first and third quartiles, whiskers extend to a maximum of 1.5 of the interquartile range and represent the minimum and maximum value within this range. **f** UMAP embedding of a subclustering of all fibroblasts from the cutaneous samples. **g** Dot plot showing canonical markers used to identify key fibroblast subsets. **h** Relative proportion of follicular dendritic cells (fDC) normalised based on all identified fibroblasts per sample (*n* = 27, coloured points). Lower and upper hinges of the boxplots correspond to the first and third quartiles, centre represents the median, whiskers extend to a maximum of 1.5 of the interquartile range and represent the minimum and maximum value within this range. Source data are provided as a Source Data file.

follow up period of 12 years. In the second case, the top expanded clone consisted mainly of non-class switched, canonical germinal centre B cells. Yet, the more abundant polyclonal plasma cells were all class switched. In routine histopathological assessments, this sample

would therefore have been classified as class switched. This highlights that the proposed practice of defining class switched and non-class switched pcMZL is not necessarily linked with a high risk of systemic disease.

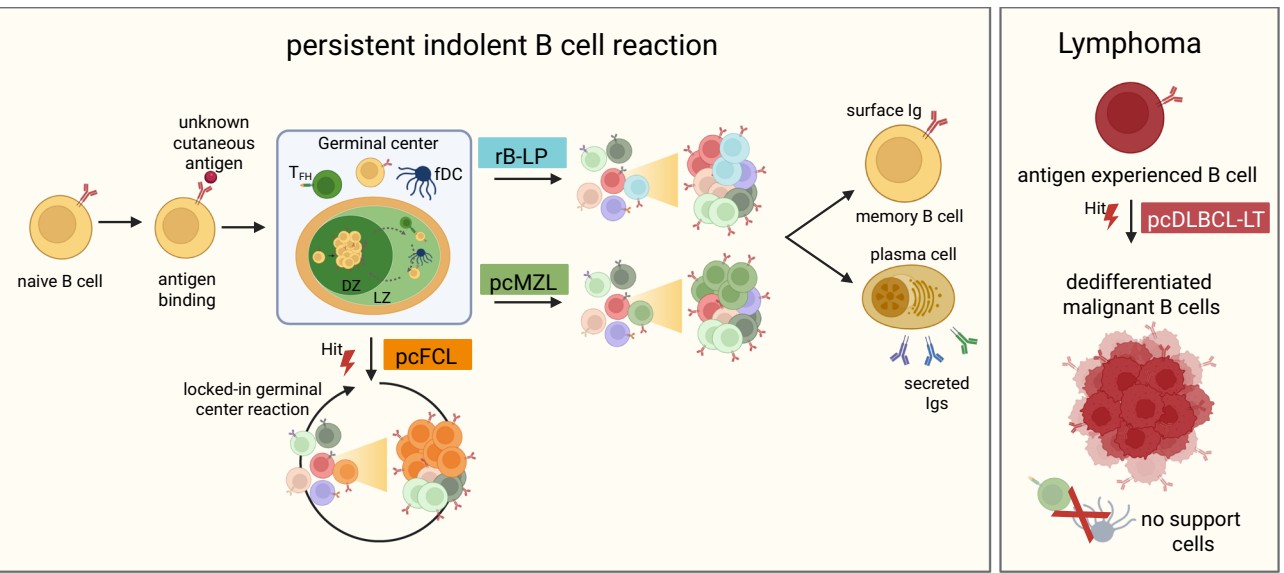

**Fig. 5 | Proposed pathophysiological mechanism of primary cutaneous B cell lymphoma subtypes.** Created in BioRender. Griss, J. (2026) https://BioRender.com/l5pft96.

In contrast to pcMZL, pcFCL exhibits a comparably high level of clonal expansion. This matches previous studies describing putative driver mutations in pcFCL[59,60]. These mutations likely arise during the ongoing germinal centre reaction, which is known to increase the risk of such genetic alterations[61]. Our data further revealed ongoing somatic hypermutation—a hallmark of the germinal centre reaction—also observed in systemic FCL (sFCL)[62]. This contrasts earlier reports where somatic hypermutation was not detected in a series of pcFCL cases[63], raising questions about whether this discrepancy stems from the higher sensitivity of our scRNA-seq approach or the existence of distinct pcFCL subsets.

Unlike sFCL, our pcFCL samples did not show signs of further differentiation but remained confined to a single phenotype. We hypothesize that while established driver mutations enable clonal expansion, sustained growth still depends on the germinal centre microenvironment. Given that this microenvironment is physiologically absent in the skin, it is plausible that pcFCL originates from a prior antigen-driven reaction. This dependence may constrain the clone to a single phenotype and contribute to the indolent behaviour of pcFCL. However, further studies are needed to fully elucidate these mechanisms.

A question we currently cannot address is whether FCL primarily occurring in the skin with bone marrow involvement is different to pcFCL. Senff et al. showed that 4.5% of patients with FCL primarily occurring in the skin had bone marrow involvement without other signs of systemic disease[64]. In their study, these nine patients showed significantly shorter survival times compared to patients with pcFCL. While in our patients CT studies showed no sign of systemic disease, bone marrow studies were not performed. Nevertheless, in our scRNA-seq data we could not identify differences between the analysed pcFCL samples and therefore believe that all our patients had pcFCL. More data is needed to assess whether cutaneous lesions arising from sFCL show different characteristics.

With respect to the staging of primary cutaneous B-cell neoplasms, our data highlights that scRNA-seq can clearly distinguish the investigated subtypes. This is in contrast to sFCL and sDLBCL where previous scRNA-seq studies showed overlapping phenotypes and high inter-patient variability[21,22]. Current recommendations for the diagnostic workup of suspected pcMZL and pcFCL patients requires at least imaging to rule out secondary lesions of systemic lymphoma[11]. While bone marrow studies are no longer recommended for pcMZL,

there is currently no consensus on its role in the workup of pcFCL patients[11]. Our data thus holds the promise that scRNA-seq of cutaneous samples may be sufficient to distinguish between primary cutaneous and systemic subtypes of B cell lymphomas arising in the skin and can act as a reference to evaluate new methods.

Current treatment strategies for indolent CBCL are primarily targeting the B cells directly. The fact that our data strongly suggests that these are antigen-driven reactions offers an avenue for new therapeutic approaches. This could open up options for potentially curative treatment approaches that eliminate the antigen and thereby stop this ongoing reaction. Nevertheless, our study is limited with respect to sample size and a demographically homogeneous cohort. Therefore, further multi-centre clinical studies are needed to validate our findings, including whether the rate of clonal expansion correlates with clinical outcome.

Finally, our data underlines the recent reclassification of pcMZL as a lymphoproliferative disorder and will need to spark a discussion on the appropriate therapeutic[65,66] and diagnostic strategies for this indolent disease.

## Methods
### Patient recruitment and sample processing
Patients were recruited at the Medical University of Vienna, Austria. Skin punch biopsies or biopsies during gastroscopy were taken after obtaining written informed consent, under a protocol approved by the Ethics Committee of the Medical University of Vienna, which includes sharing of patient characteristics (EK 1360/2018). As per the regulations of the Ethics Committee, patients were only compensated for additional time required for the study participation. Histopathological diagnoses were confirmed by two independent board-certified histopathologists. All patients with pcMZL, pcFCL, and pcDLBCL-LT were further staged using full body CT scans to rule out systemic disease. For all patients with rB-LP, borrelia infection was ruled out. After processing using the Skin Dissociation Kit by Miltenyi Biotech (Bergisch Gladbach, Germany)[67–69], single-cell suspensions were subjected to scRNA-seq using the Chromium Single Cell Controller and Single Cell 5' Library & Gel Bead Kit v2 (10X Genomics, Pleasanton, CA) according to the manufacturer's protocol. B cell receptor (BCR) sequences were enriched from cDNA using the VDJ Kit workflow by 10X Genomics. Sequencing was performed using the Illumina NovaSeq platform and the 150 bp paired-end configuration.

## Data analysis

Raw data from scRNA-seq was preprocessed using Cell Ranger version 7.0.1 invoking the command "multi" and aligned to the human reference genome assembly "refdata-gex-GRCh38-2020-A" and Cell Ranger V(D)J segment reference "refdata-cellranger-vdj-GRCh38-alts-ensembl-7.0.0".

IgBLAST[42] v1.20 was used on the BCR filtered contigs to quantify the somatic hypermutation (SHM), presented as *"1 - identity in variable region"*.

B cell clones were identified from the BCR data through the following steps. Cells featuring more than two contigs or multiple contigs for the same BCR chain (heavy or light chain), and clones with more than two CDR3 chains, were removed. Subsequently, if one of the two most frequent clones only contained one heavy or one light chain were merged with clones were both chains were identified if all cells shared the same V, C, and J genes as the evaluated clonotype, but only if all cells displayed at least 85% sequence identity in both light and heavy CDR3 sequences compared to the reference sequence of the evaluated clonotype, using global pairwise alignment from Biostrings[70] R package v2.68 (BLOSUM100 matrix with a gap opening and gap extension penalty values of 10 and 4, respectively).

Every sample was processed using R (version 4.1.0) and Seurat (version 5.0.1)[71]. Ambient RNA was removed using DecontX[72] and doublets removed using scDblFinder[73]. During the quality assessment, only cells with at least 500 genes and a maximum of 10%, 5% and 1% of mitochondrial, haemoglobin and platelet genes, respectively, were kept for downstream analysis.

Afterwards, samples were normalised and scaled regressing for percentage of mitochondrial genes and sample origin. 30 dimensions were used when invoking the Seurat functions RunPCA, FindNeighbors and RunUMAP. Subsequently, samples were integrated using the FastMNN method[74]. Leiden algorithm was applied on the shared nearest neighbour graph clustering prior to cell annotation as suggested by Heumos et al.[75]. In order to achieve a deeper annotation, B and T cells were subclustered, variance was stabilised using SCTransform v2[76] and ScaleData, respectively, with 15 and 20 dimensions, respectively, for subclusterings. Cell interaction analysis was performed using CellChat v2.2[77]. Differential expression analysis using Seurat's FindMarkers function with default parameters. Genes were filtered based on FDR ($\leq$0.01) and an expression of 20% within the cells of the respective group (pct.1 and pct.2 for up- and down-regulated genes, respectively).

All *p*-vaues reported in the manuscript were corrected for multiple testing using the Benjamini–Hochberg ("FDR") correction.

## Immunohistochemistry

Multiplex immunostainings were conducted as previously described[78]. Briefly, 4 µm sections were deparaffinized and antigen retrieval was performed in heated citrate buffer (pH 6.0) and/or Tris-EDTA buffer (pH 9) for 30 min. Thereafter, sections were fixed with 7.5% neutralised formaldehyde (SAV Liquid Production GmbH, Flintsbach am Inn, Germany). Each section was subjected to 6 successive rounds of antibody staining, consisting of protein blocking with 20% normal goat serum (Dako, Glostrup, Denmark) in PBS, incubation with primary antibodies, biotinylated anti-mouse/rabbit secondary antibodies and Streptavidin-HRP (Dako), followed by TSA visualisation with fluorophores Opal 520, Opal 540, Opal 570, Opal 620, Opal 650, and Opal 690 (PerkinElmer, Waltham, MA, USA) diluted in 1X Plus Amplification Diluent (PerkinElmer), Ab-TSA complex-stripping in heated citrate buffer (pH 6.0) and/or Tris-EDTA buffer (pH 9) for 30 min, and fixation with 7.5% neutralised formaldehyde. Thereafter, nuclei were counterstained with DAPI (PerkinElmer), and sections were mounted with PermaFluor fluorescence mounting medium (Thermo Fisher Scientific, Waltham, MA, USA). Multiplexed slides were scanned on a Vectra Multispectral Imaging System version 2 following the manufacturer's protocol (InForm 2.4, Perkin Elmer). All phenotyping and subsequent quantifications were performed blinded to the sample identity using QuPath version 0.5.1[79].

## Statistics and reproducibility

All data presented in this manuscript was generated from primary patient samples. Due to limited sample availability technical replicates were not performed. Throughout all analyses, all available samples were used. For scRNA-seq, these were 9 pcMZL, 5 pcFCL, 4 pcDLBCL-LT, 5 cutaneous rB-LP, and 4 samples from healthy volunteers. For IHC analyses, these included 4 rB-LP, 16 pcMZL, 13 pcFCL, and 7 pcDLCBL-LT samples.

## Reporting summary

Further information on research design is available in the Nature Portfolio Reporting Summary linked to this article.

## Data availability

The processed scRNA-seq data generated in this study have been deposited in the GEO database under accession code GSE218861. Data from healthy control samples are available on GEO under GSE173205[80]. The raw scRNAs-seq data are protected and are not available due to data privacy laws. The CBCL scRNA-seq data used in this study are available in the European Genome-Phenome Archive (EGA) database under accession code EGAD00001006829. The sFCL scRNA-seq data used in this study are available in the EGA database under accession code EGAS00001006052. The sDLBCL scRNA-seq data used in this study are available in the GEO database under accession code GSE182436. Source data are provided with this paper.

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

## Acknowledgements

This work was funded by research grants from the Austrian Science Fund to P.M.B. (grant number KLI 849-B) and J.G. (grant number P35937). S.N.W. was supported by research grants from the Austrian Science Fund (grant number P31127 and IPPTO project number DOC 59-B33). I.O. was supported by a DOC fellowship from the Austrian Academy of Science (Grant number 27228). The Vienna Scientific Cluster (Project No. 71839) is gratefully acknowledged for providing computational resources.

## Author contributions

Designed research J.G., C.J., S.N.W., P.M.B. Sample acquisition J.G., C.J., P.M.B., S.P., W.D. Histopathological analysis M.D., I.S.K. Sample analysis L.S., U.M., M.F., B.A., B.M.L., M.S., S.Z.S., C.W., S.N.W., W.W. Data analysis J.G., S.G., I.O., V.N. Acquisition of funding J.G., P.M.B. Writing of manuscript J.G., C.J., P.M.B., S.N.W., B.M.L.

## Competing interests

J.G. received personal fees from AbbVie, Eli Lilly, Pfizer, Boehringer Ingelheim and Novartis. C.J. has received personal fees from Boehringer Ingelheim, LEO, Pfizer, Recordati Rare Diseases, Eli Lilly, Novartis, Takeda, Kyowa Kirin, STADA, UCB, BMS, AbbVie, Janssen, Stemline, and Almirall. C.J. is an investigator for Eli Lilly, Novartis, AbbVie, Boehringer Ingelheim, Incyte, 4SC, and Innate Pharma. W.W. has received personal fees from LEO Pharma, Pfizer, Sanofi Genzyme, Eli Lilly, Novartis, Boehringer Ingelheim, AbbVie, and Janssen. W.D. has received personal fees from Boston Scientific, Olympus, Medtronic, Norgine, MSD, Takeda and Ferring. P.M.B. has received personal fees from Almirall, Sanofi, Janssen, Amgen, LEO Pharma, AbbVie, Pfizer, Boehringer Ingelheim, GSK, Regeneron, Eli Lilly, Celgene, Arena Pharma, Novartis, UCB Pharma, Biotest and BMS. P.M.B. is an investigator for Pfizer and Abbvie.
