## [Transparent Peer Review file · Nature Communications]

Indolent primary cutaneous B-cell lymphomas resemble persistent antigen reactions without signs of dedifferentiation

Corresponding Author: Dr Johannes Griss

Version 0:

Reviewer comments:

Reviewer #1

(Remarks to the Author)

This study presents a highly detailed single-cell RNA sequencing (scRNA-seq) analysis of primary cutaneous B-cell lymphomas (CBCLs), specifically pcMZL, pcFCL, and pcDLBCL-LT. It highlights new insights into their clonal evolution, transcriptomic profiles, and B-cell receptor (BCR) characteristics. The major claim that pcMZL behaves more like a lymphoproliferative disorder rather than a true lymphoma is potentially paradigm-shifting and offers novel perspectives on diagnostics, classification, and therapeutic considerations.

Key strengths that justify publication if revised appropriately:

Novelty & Importance – This study fills a gap in molecular profiling of CBCLs, a rare lymphoma subtype.

Methodological Rigor – The use of scRNA-seq, BCR sequencing, and multiplex immunohistochemistry (IHC) provides a comprehensive multi-modal analysis.

Well-Structured & Logical Flow – The manuscript is well-organized, with a clear hypothesis and stepwise data presentation.

Potential Clinical Impact – The reclassification of pcMZL as a lymphoproliferative disorder could influence future lymphoma classification and management.

The study provides strong transcriptomic insights but lacks a direct comparison to existing diagnostic standards. The clinical consequences of pcMZL reclassification need to be clarified.

Provide a comparison of scRNA-seq to current diagnostics (Page 3, Lines 20-25): compare scRNA-seq vs. IgH rearrangement PCR vs. IHC vs. flow cytometry in terms of diagnostic accuracy; include specificity and sensitivity values where possible.

Please, discuss how scRNA-seq findings align with WHO and ICC lymphoma classification criteria and clarify how findings will impact patient management (Page 16, Lines 10-15):

1. Would patients previously diagnosed with pcMZL now be managed differently?
2. Should routine scRNA-seq be considered for diagnosis or prognosis?

Can you make therapeutic implications (Page 21, Lines 5-10) by discussing whether pcMZL patients with high clonal expansion might require targeted therapies or more aggressive treatment?

It would be interesting to explore potential immunotherapy targets identified from transcriptomic data.

Showing clinical applicability will be beneficial and will make findings more translational and relevant to Nature Communications readers.

It is a common problem with rare diseases to work with the limited patient cohort (pcMZL n=9, pcFCL n=5, pcDLBCL-LT n=4, rB-LP n=5), the common issue which limits generalizability. In this particular study, the findings are not tested in an independent validation cohort and IHC validation (n=19) is not large enough for meaningful conclusions.

It is recommended to expand validation using publicly available datasets (Page 7, Lines 4-10):

Validate scRNA-seq signatures against GEO datasets (e.g., GSE218861, EGAS00001006052, EGAS00001004904) and to perform bulk RNA-seq validation on an additional independent cohort.

Additional technical validation (Page 10, Lines 3-6): the investigators can use spatial transcriptomics to confirm scRNA-seq findings at the tissue level. In addition, multiplex flow cytometry can validate protein expression of key markers in fresh

patient samples.

Regarding statistical robustness (Page 14, Figure 4A), the investigators should consider applying Bonferroni or Benjamini-Hochberg corrections to prevent false positives in differential expression analysis and to use hierarchical mixed-effects models to account for patient-to-patient variability.

Addressing these concerns would make findings more robust.

I would recommend improving figures and supplementary data integration including clarifying the labels and legends, improving references to the supplementary figures in the main text, and overall improved data presentation (it is not always intuitive for a non-specialist). For example, I would recommend to improve clarity in key figures (Page 8, Figure 2A): add axis labels, clustering annotations, and improved color contrast; provide UMAP embeddings with more detailed explanations. Please, make sure that all supplementary figures are directly linked to main text (Page 6, Figure 1) by explicitly referencing which supplementary figures support key claims. Also, ensure figure legends provide enough information to be understandable on their own. It would be helpful to add summary diagrams (Page 12, Lines 18-22), such as a schematic of B-cell evolution in pcMZL vs. rB-LP would help non-specialists grasp findings more easily. Enhancing readability, accessibility, and figure integration, will make results easier to interpret.

It is important to acknowledge study limitations and to justify cohort constraints. Currently, the study does not adequately acknowledge limitations of sample size, demographic bias, and antigen-driven hypotheses. For example, the antigen-driven persistence model is not directly tested, so that suggesting future multi-center studies to validate across diverse cohorts is warranted. To address antigen-driven persistence limitations (Page 15, Lines 8-12), functional validation studies to confirm antigen-driven B-cell expansion should be considered. Furthermore, peptide screening assays to identify putative antigens involved in pcMZL would be of great interest, though might be beyond the scope of this study. Also, longitudinal studies tracking B-cell evolution in pcMZL patients over time might be considered as future directions (Page 24, Conclusion).

Reviewer #2

(Remarks to the Author)

This paper tackles the interesting challenge that we have in hematopathology on how to distinguish reactive lymphoid lesions harboring clonal populations from truly neoplastic lesions, and when one should use the term "lymphoma." There are several lymphoproliferative disorders that have clonal B cell populations, but are clearly not malignant, and we, as a field struggle with how to deal with these. This is certainly an important question and one of clinical impact.

Griss et al ambitiously tackle this problem using single cell sequencing of several of these relatively rare lesions, and report some interesting findings and make some valid arguments about whether particular lesions may be reactive or neoplastic on the basis of the dominance of the main clone, whether the lesion contains "aberrant cells" and whether there was persistence of germinal center reaction. The authors illustrate that our routine approaches in pathology to distinguish lymphoma or not are quite flawed. This was best illustrated by the two examples of the switched and non-switched pcMZL where the authors demonstrate that this criteria is not sufficient to distinguish a lymphoma from a reactive process.

While the manuscript and the data in it were quite interesting, the findings remain mostly descriptive, and are not sufficient to make the broad definitive conclusions that the authors make about these disease entities. There are a number of weaknesses along these lines:

1. Given the amount of heterogeneity present in these lesions, the number of samples analyzed for each entity is very small. In addition, the authors do not describe how tumors are sampled, and there is no consideration regarding sampling variation, which would impact the % of a clonal population present and the % of supportive cells present, for example. It would be nice to see a few examples of multiple sampling from one tumor to assess the level of variability that may be present just in sampling different areas.
2. These are challenging entities to diagnosis and nomenclature has changed over time. There is no description of how the diagnoses were verified, pathology review, and validation on how the patients were clinically staged. There is limited clinical data, survival data, or consideration of age/gender, etc. This is critical information needed if one will make definitive statements about particular entities.
3. I think that the way different B cell populations are defined are limited and use minimal markers. For example, aberrant populations seem to be mostly defined by having variation from the normal expression of 2-3 markers.
4. Independent validation of expression data was quite limited
5. Assessment of data QC seemed quite limited

For all of these reasons, the data, while interesting are not, in my opinion, sufficient to support the broad claims made about pcMZL, pcFL, etc.

Reviewer #3

(Remarks to the Author)

The authors have generated and analyzed one of the largest single cell RNA-Seq datasets generated for cutaneous B cell lymphomas. These include pcMZL n=9, pcFCL n=5, pcDLBCL-LT n=4, rB-LP n=5. They do standard informatic analyses to

cluster the cell types, to identify the putative clone, and to elucidate the putative cell of origin.

The field of cutaneous B cell lymphomas is a burgeoning field with new insights that can have broad implications for 1) cutaneous lymphomas, 2) cutaneous immunology, and 3) B cell biology. They share some markers about the phenotypic diversity in pcMZL that contrasts with the more uniform distribution of the cells in the other cutaneous lymphomas.

major points:

The conclusions from this dataset are limited, which I will discuss here.

1. One of the major questions in the field is how cutaneous and systemic lymphomas are different? Why are the cutaneous B cell lymphomas so indolent? The description of the differences between the systemic and cutaneous lymphomas is inconsistent. A fundamental feature of cutaneous lymphomas is that they are genetically and proteomically distinct from the systemic lymphomas. How much of this can be attributed to the differences in the cell of origin? Restricting much of the analyses to skin lymphomas and skin controls limits the impact. Are the pcDLBCL-LTs a different cluster because the appropriate cells of origin, ABC-type cutaneous lymphomas, are not represented in this dataset? or because they are truly different?

2. Phenotypic features that distinguish cutaneous and systemic lymphomas are almost completely ignored. Many of these are derived from genetic differences between the cutaneous and systemic counterparts. These fundamental biological insights are not addressed. For example, pcFCL have significantly less BCL2 expression than their systemic counterparts. These comparisons are required to make the analyses interesting.

3. The use of cutaneous controls is unusual. While there are potential lymphoid aggregates in the skin; traditionally, many of these germinal zone aggregates are formed in the lymph node. It would be better to compare to see if the cells here are similar to the ones found in lymph nodes or different. If so, how?

4. There is almost no discussion about the microenvironment re: signals between pro-tumor and anti-tumor cells with the lymphoma cells. A wealth of data suggest that many mutations in these CBCLs occur in cell surface proteins that are predicted to mediate interactions with the environment, such as TNFRSF14, FAS, and/or MHC class I. These are not addressed.

5. These tumors are rare, meaning that it is difficult to expand the dataset to ensure generalizability. In that setting, the manuscript will require orthogonal validation through spatial and/or surface IHC to describe how these markers can be leveraged across larger datasets.

Minor.

1. The manuscript is written in a way that makes it difficult to read for non-experts in the field. Tables that describe the differences between clinical entities and their systemic counterparts would make it easier to read.

2. Genetic information can be gleaned from well curated datasets. Are any mutations observed in these tumors?

3. The choice of controls such as MALT lymphomas are hard to understand compared to other systemic lymphomas.

4. Description of the "bystander" B cells would be useful. Are these cells defined by merely the absence of the clonal BCR? if so, could this be a technical error? do they express alternative BCRs?

Version 1:

Reviewer comments:

Reviewer #1

(Remarks to the Author)

See attached file

Reviewer #2

(Remarks to the Author)

The study tackles the challenging area of how to distinguish indolent primary cutaneous lesions and how we should define malignant lymphomas in this setting. The authors have done this through detailed single-cell RNA sequencing analysis of these primary cutaneous lymphomas. Though the paper as originally submitted was definitely intriguing, this author identified a number of weaknesses, including the small n size for the conclusions being made, no independent validation, description of how these cases were classified histologically, limited phenotypic characterization, description QC metrics.

The authors have since added new and informative samples and phenotypic and other analyses that address the concerns raised previously. The paper is strong and provides very intriguing evidence that these indolent primary lymphomas are locally driven and provide a basis for future discussion regarding how these should be classified (probably not as malignant lymphomas). However, I have conceptual concern that I believe should be addressed:

Much of the evidence that pcMZL and pcFCL may be a locally driven process (and perhaps not truly represent malignant lymphoma) rests in the fact that pcMZL, pcFCL exhibit persistent germinal center reaction, not observed in pcDLBCL and gastric MALT. This represents very strong evidence that pcMZL (especially with the relatively small clonal B populations seen in pcMZL) represents a localized lymphoproliferative disorder and not a true lymphoma.

However, for pcFCL where you have shown larger relative clonal populations, it seems as if there is not sufficient evidence to make the same statement. I wonder about the comparison of pcFL to sFL. FL is a systemic disease and considered malignant, but it also maintains ongoing somatic hypermutation, and also harbors the important elements of the microenvironment, on which these cells are thought to depend. Unlike pcMZL, your data shows that pcFL has the second highest level of ongoing SH with a narrow range (similar to pcDLBCL). It also seems to have a higher clonal percentage. Therefore, the distinction between sFL and pcFL on the parameters that you are using to reclassify pcMZL is less clear.

I think this should be addressed in the discussion by either 1) providing more evidence or further explaining the distinction between pcFCL and sFL in terms of ongoing SH, clonal populations, microenvironment - or restricting the main hypothesis about the biology of primary cutaneous lymphomas to pcMZLs, and acknowledging that pcFL may represent a neoplastic process whose anatomic restriction remains less understood.

To this end, I would suggest modifying the title to focus on "pcMZL" and not generally primary cutaneous lymphomas.

Overall, this is an excellent thought-provoking manuscript that provides new data to help us better understand these primary cutaneous lesions.

Reviewer #3

(Remarks to the Author)

The authors present a minimally revised manuscript.

My concerns have not been adequately addressed. I will again list them here. In sum, the inability to attempt these analyses suggest a broad ignorance of the field, the underlying B cell lymphoma biology, the techniques accessible to them, and the statistical rigor that additional analyses would provide. The points outlined below are not intended to be complete but rather to highlight the overconfidence by which the authors tried to address scientific points raised by myself and the other reviewers.

1. The fields of B cell biology already capture parallels between normal B cell biology and B cell lymphomas. This observation does not make their paper impactful. In fact, these descriptions and analogies are baked into their disease entities. The follicular, marginal zone, and diffuse B cell lymphomas have many clear analogies to B cells in different phases of differentiation and expansion. These cells of origin are, in fact, even more detailed as the diffuse large B cell lymphomas comprise multiple clinically relevant phenotypic differences between activated B cell type and germinal cell B cell type. Moreover, there are additional differences amongst the ABC-type leading to clinically relevant differences between nodal and extra nodal where these can be broken down to systemic ABC-type DLBCL, CNS/test ABC-type DLBCL +/- skin, and mediastinal DLBCL. These have now been distributed across additional subtypes based on genetics and transcriptional programs. They have clear differences that can be identified across disease subtypes and across cells-of-origin.

I and the other reviewers had suggested a careful and thoughtful consideration of these differences to highlight what is truly novel biology unlocked by their single cell RNA-Sequence analyses. These are inadequately covered by the new Figure 3. This should have been thoughtfully done with known markers across disease subtypes and in fact belies an insufficient understanding of the field.

2. The differences from tumor to tumor and across subtypes strongly correlate with genetics, suggesting a determinative relationship between mutations and phenotypes. Cancers are subject to both copy number and point mutations. There are known mechanisms to identify putative copy number mutations and/or point mutations across single cell RNA-seq datasets. While the authors may not have initially intended to analyze these, their disregard for this suggestion suggests that they under appreciate the importance of genetics to lymphoma biology, which makes it difficult to appreciate whether they understand the field at all and/or the bioinformatics tools that they are deploying.

For example, again, there are canonical DLBCL mutations in systemic lymphomas, in skin lymphomas, and shared across the two. There are similar mutations in the other cutaneous B cell lymphoma subtypes.

3. I have not reviewed or read a lymphoma manuscript with single cell RNA-Seq that has not described the host response to the tumor. CellChat does not describe the tumor microenvironments. It does not describe the cellular composition of the lymphomas across patients and within individual patients.

The tumors themselves have genetically-encoded differences in immune ligands. These are not addressed in a systematic way. Their de-prioritization based on CellChat does not mean that they have been actively down regulated by the tumor and/or if this has effects on other cells in the microenvironment. The decision not to pursue this suggests an inadequate appreciation for what tumor immunology represents.

4. The generalizability of these findings. These studies are the smallest genomic-clinical cohorts that I have seen. The studies with bulk RNA or DNA-Seq have much higher samples which enable broader generalizations of their data. Since these are rare tumors, we had suggested spatial transcriptomics to confirm orthogonally these findings, identify novel spatial

interactions, and enable extension of their work to new samples for which fresh single cells are not available. Their claims that this would limit the ability to track antigen-responsive cells (Especially for matched tumors) reflects an ignorance about the potential of integrating these datasets. Again, this highlights an overconfidence in their ability to make conclusions based on their limited understanding of the lymphoma biology, the technical approaches, and the statistical rigor.

Version 2:

Reviewer comments:

Reviewer #2

(Remarks to the Author)

As mentioned previously, the data in this paper are interesting and support the idea that pcMZL may be a localized antigen-driven lymphoproliferative disorder as opposed to a true lymphoma. However the evidence in the manuscript that the same may be claimed for pcFL is much weaker in comparison. My suggestion was either to provide more evidence supporting this claim for pcFL – or restricting the claim for primacy cutaneous MZL both in the discussion in the title.

The authors' response is insufficient and presents the data in the manuscript and hypotheses based on this data - in a vacuum, totally ignoring what is known in the field about cutaneous and systemic lymphomas:

"In contrast to nodal FCL the support cells required for SH are physiologically not present in skin".

- It is actually well known that many of the microenvironmental cells relevant to pcFL are normally present in the skin, though with different frequency and organization, including CD4+ T cells, CD8+ cytotoxic T cells, dermal dendritic cells, macrophages and stromal cells. While Tfh cells are rare in skin, they can be recruited during inflammation.

"A true lymphoma such as pcDLBCL-LT therefore lacks these support cells as it is unable to recruit them (in contrast to nodal DLBCL where these support cells are already present)."

- This statement does not make sense. Both pcDLBCL-LT and nodal DLBCL are characterized by sheets of large neoplastic and are independent of microenvironment support cells (for the most part). Both pcDLBCL-LT and nodal DLBCL are both malignant neoplasms due to their genetic alterations and obviously does not need to recruit support cells. The differences between these lymphomas do not lie in difference in ability to recruit support cells, but major differences in genetic alterations specific for each entity. These genetic differences are very well described.

"We therefore feel that our hypothesis that pcFCL originates from an initial antigen-driven response is supported (yet not proven) by our data and a more likely explanation than altered B cells that are able to recruit the complete environment that provides necessary survival signals without any signs of systemic disease."

-- This statement is also problematic.

1) It is still not clear what component of the current data supports the idea that pcFCL originates from an initial antigen-driven response. That data is lacking.

2) Stating that an entity originates from an initial antigen-driven response is very different than saying an entity is an antigen-driven lymphoproliferative process that is not lymphoma. Gastric MALT - which you correctly describe in the manuscript as true lymphoma originates from an antigen-driven response – namely H.pylori. Sometimes, these lymphomas remain responsive to H.pylori eradication, and sometimes they become independent of H.pylori stimulation as a result of genetic alterations. So whether pcFL initially originates from an antigen-driven response is sort of irrelevant to whether it should be called a lymphoma.

3) Unfortunately, the authors present their work in a vacuum.

- There are two current classifications in lymphoma – WHO and ICC (International Consensus Classification). Although the WHO5 continues to classify pcMZL as a lymphoma, the 2022 ICC has already been downgraded it to a persistent antigen-driven "lymphoproliferative disorder" based on current literature, clinical outcomes, association with *Borrelia burgdorferi*, – so this concept is not new. This should be mentioned in the manuscript. The current data mainly serves to support this point.

- There is some suggestion in the literature that pcFL also originates from an antigen-driven response, but no great definitive evidence – neither is there in this manuscript. However, there is clear evidence that pcFL exhibits some of the features of a lymphoma – including for examples its tendency to recur locally after excision, and predominance of clonal cells within the lesions. That being said, it's not as aggressive as pcDLBCL because it tends to stay in the skin – even when it recurs, and prognosis is excellent, while pcDLBCL has poor prognosis.

We fully share the reviewer's view that our data is less clear on pcFCL compared to pcMZL. We tried to highlight this in Figure 5 where we tried to clearly show that we do believe that pcFCL cells do acquire some kind of genetic alteration that causes their proliferation.

- There are papers that demonstrate the phenotypic (BCL2 expression) and genetic alterations (CBP, MLL2, KTM2D, etc) mutations that underlie differences between pcFL and systemic FL that infiltrate skin – this should not be ignored in the manuscript as part of the discussion!!

"As it was not our intention to present the evidence for pcFCL at the same level as pcMZL, we adapted our discussion as

follows: ... Yet, these clones were still undergoing somatic hypermutation as part of the germinal centre reaction, which nevertheless is also observed in sFCL. Thereby, the clone may be able to expand, yet still requires the support from the germinal centre environment. As this microenvironment is physiologically not present in skin, we believe that it is most likely that pcFCL develops from a previous antigen-driven reaction. This dependence locks it in its single phenotype and may explain pcFCL's indolent behaviour. Nevertheless, further studies are needed to determine whether pcFCL should be classified as lymphoma or lymphoproliferative reaction."

To this reviewer, this explanation is unclear (I am not sure what author is trying to communicate) and ignores much of what is already in the literature on this topic, as described above

- PMID: 40227781
- PMID: 33045225
- PMID: 33560380

I would advise authors to really integrate what has already been published in the field, and to focus your conclusions on pcMZL as being not lymphoma - and not pcFL.

Version 3:

Reviewer comments:

Reviewer #2

(Remarks to the Author)

The authors have addressed questions satisfactorily

Manuscript number: NCOMMS-24-82894A

Dear Editors,

Thank you very much for your invitation to resubmit our manuscript. We would like to thank all the reviewers for their comments, which we believe have significantly helped to improve the manuscript. Please find below our point-by-point responses to those comments, and we hope that these additions and corrections to the manuscript now warrant publication in your journal.

Reviewer #1 (Remarks to the Author):

This study presents a highly detailed single-cell RNA sequencing (scRNA-seq) analysis of primary cutaneous B-cell lymphomas (CBCLs), specifically pcMZL, pcFCL, and pcDLBCL-LT. It highlights new insights into their clonal evolution, transcriptomic profiles, and B-cell receptor (BCR) characteristics. The major claim that pcMZL behaves more like a lymphoproliferative disorder rather than a true lymphoma is potentially paradigm-shifting and offers novel perspectives on diagnostics, classification, and therapeutic considerations.

Key strengths that justify publication if revised appropriately:

Novelty & Importance – This study fills a gap in molecular profiling of CBCLs, a rare lymphoma subtype.

Methodological Rigor – The use of scRNA-seq, BCR sequencing, and multiplex immunohistochemistry (IHC) provides a comprehensive multi-modal analysis.

Well-Structured & Logical Flow – The manuscript is well-organized, with a clear hypothesis and stepwise data presentation.

Potential Clinical Impact – The reclassification of pcMZL as a lymphoproliferative disorder could influence future lymphoma classification and management.

The study provides strong transcriptomic insights but lacks a direct comparison to existing diagnostic standards. The clinical consequences of pcMZL reclassification need to be clarified.

Provide a comparison of scRNA-seq to current diagnostics (Page 3, Lines 20-25): compare scRNA-seq vs. IgH rearrangement PCR vs. IHC vs. flow cytometry in terms of diagnostic accuracy; include specificity and sensitivity values where possible.

Response: We very much thank the reviewer for these suggestions!

As suggested, we added an in-depth comparison between cutaneous and systemic lymphoma. These also showed the currently used difference in BCL6 and BCL2 expression between pcFCL and sFCL.

We additionally did try to perform paraffin-based BCR sequencing using Thermo Fischer's IonTorrent system. This is the diagnostic gold-standard in our hospital. Yet, out of the 12 processed samples, despite sufficient DNA, 8 failed to unclear technical issues. Out of the 4, two were deemed as

polyclonal by the IonTorrent system, while the scRNA-seq data showed a clonal expansion of 50%. Based on these results the Department of Molecular Pathology now stopped the use of the IonTorrent system for paraffin-based analyses and is working with the manufacturer to analyse these issues.

In contrast to cutaneous T cell lymphoma, there is currently no gold standard to diagnose CBCL using flow cytometry.

Finally, our scRNA-seq sample set is too small to perform robust sensitivity and specificity analyses. We are currently working on different clinically usable assays that could be used on larger sample sets and subsequently be fit for routine clinical use. Nevertheless, these efforts are beyond the scope of this study.

Please, discuss how scRNA-seq findings align with WHO and ICC lymphoma classification criteria and clarify how findings will impact patient management (Page 16, Lines 10-15):

1. Would patients previously diagnosed with pcMZL now be managed differently?

As a first consequence patients with non-class switched pcMZL are no longer staged using CT scans at our hospital. We hope that our data will help to settle the current discussion on whether class-switched cases of pcMZL are a true lymphoma or not.

Furthermore, we very much hope and believe that our data will nevertheless spark a wider discussion into which type of treatments are warranted for the indolent types of CBCL.

Accordingly, we very strengthened these points in the discussion as follows:

We therefore, for the first time, present molecular evidence that aligns with the concept that pcMZL is a lymphoproliferative disorder and not a true lymphoma.

2. Should routine scRNA-seq be considered for diagnosis or prognosis?

We believe that we were able to show that scRNA-seq can unambiguously differentiate between these entities. It would therefore definitely be advantageous to at least use such kind of data as a gold standard for studies to circumvent current diagnostic challenges.

We highlighted this in our discussion as follows:

Our data thus holds the promise that scRNA-seq of cutaneous samples may be sufficient to distinguish between primary cutaneous and systemic subtypes of B cell lymphomas arising in the skin.

Can you make therapeutic implications (Page 21, Lines 5-10) by discussing whether pcMZL patients with high clonal expansion might require targeted therapies or more aggressive treatment?

We are yet unable to make this statement based on our cohort as we did not have any patient with pcMZL that showed a high level of clonal expansion. Nevertheless, we strongly believe that we need to characterise patients with non-class switched pcMZL more thoroughly. Our data shows that the class-switch alone does not sufficiently identify patients with aggressive diseases.

In our dataset, only samples from pcDLBCL-LT showed a clonal expansion similar to systemic lymphomas. These patients are already receiving aggressive therapies.

We nevertheless do believe that our data offers an avenue for new treatment approaches, as highlighted below.

It would be interesting to explore potential immunotherapy targets identified from transcriptomic data. Showing clinical applicability will be beneficial and will make findings more translational and relevant to Nature Communications readers.

Our data fully supports the current notion that indolent CBCL do not require aggressive treatment. We believe that our data primarily suggest that treatment strategies should focus on identifying the underlying antigen and potentially eliminate it to interrupt the ongoing germinal center reaction.

We explicitly added this to our discussion with a new paragraph:

Current treatment strategies for indolent CBCL are primarily targeting the B cells directly. The fact that our data strongly suggests that these are antigen-driven reactions offers an avenue for new therapeutic approaches. Subsequent studies are needed to functionally validate this finding and identify these putatively driving antigens. This could open up options for potentially curative treatment approaches that eliminate the antigen and thereby stop this ongoing reaction.

It is a common problem with rare diseases to work with the limited patient cohort (pcMZL n=9, pcFCL n=5, pcDLBCL-LT n=4, rB-LP n=5), the common issue which limits generalizability. In this particular study, the findings are not tested in an independent validation cohort and IHC validation (n=19) is not large enough for meaningful conclusions.

In order to strengthen our cohort, we were able to more than double the number of samples used for IHC validation (now n = 40). All of these new samples matched the findings from our scRNA-seq cohort.

Moreover, we were able to acquire two new samples from a patient with extensive rB-LP to show that the lesions are similar both with respect to clonal expansion and phenotypic composition. Finally, we added a separate analysis of two patients where we acquired samples within 6 and 12 months respectively that also showed that the disease remained unchanged.

Thereby, we believe that we could arrive at a relevant sample size and further support the robustness of our methods.

It is recommended to expand validation using publicly available datasets (Page 7, Lines 4-10): Validate scRNA-seq signatures against GEO datasets (e.g., GSE218861, EGAS00001006052, EGAS00001004904) and to perform bulk RNA-seq validation on an additional independent cohort. Additional technical validation (Page 10, Lines 3-6): the investigators can use spatial transcriptomics to confirm scRNA-seq findings at the tissue level. In addition, multiplex flow cytometry can validate protein expression of key markers in fresh patient samples.

As suggested we added an in-depth analysis and comparison to our data from the only available scRNA-seq dataset on primary cutaneous B cell lymphoma (Ramelyte *et al.*, new Figure 1H). These samples fully matched our own analysis.

Additionally, we were able to analyse a matched sample from one of our pcDLBCL-LT patients that showed a large number of presumably CD27+ B cells. This was fully confirmed by our IHC analysis further strengthening the accuracy of our scRNA-seq analysis (new Figure 2).

Regarding statistical robustness (Page 14, Figure 4A), the investigators should consider applying Bonferroni or Benjamini-Hochberg corrections to prevent false positives in differential expression analysis and to use hierarchical mixed-effects models to account for patient-to-patient variability. Addressing these concerns would making findings more robust.

All p-values presented in the manuscript were corrected for multiple testing using the Benjamini-Hochberg ("FDR") correction. This is now clearly stated throughout the manuscript when the respective p-values are reported. Additionally, we adapted the methods section accordingly:

All p-values reported in the manuscript were corrected for multiple testing using the Benjamini-Hochberg ("FDR") correction

I would recommend improving figures and supplementary data integration including clarifying the labels and legends, improving references the supplementary figures in the main text, and overall improved data presentation (it is not always intuitive for a non-specialist). For example, I would recommend to improve clarity in key figures (Page 8, Figure 2A): add axis labels, clustering annotations, and improved color contrast; provide UMAP embeddings with more detailed explanations.

As suggested we re-created all supplementary figures and considerably extended the respective figure legends. We very much hope that these are now clearly tied to the main manuscript's text.

In the main manuscript, original figures 2 and 3 were removed and replaced with a new figure showcasing the analysis of our large IHC validation cohort (new Figure 2), as well as the in-depth comparison against systemic lymphoma (new Figure 3). Additionally, we added more graphical summaries to improve the presentation for non-specialists.

Please, make sure that all supplementary figures are directly linked to main text (Page 6, Figure 1) by explicitly referencing which supplementary figures support key claims. Also, ensure figure legends provide enough information to be understandable on their own. It would be helpful to add summary diagrams (Page 12, Lines 18-22), such as a schematic of B-cell evolution in pcMZL vs. rB-LP would help non-specialists grasp findings more easily. Enhancing readability, accessibility, and figure integration, will make results easier to interpret.

As suggested we adapted all supplementary figures and their legends. Additionally, we added the suggested graphical summary to the new Figure 3.

It is important to acknowledge study limitations and to justify cohort constraints. Currently, the study does not adequately acknowledge limitations of sample size, demographic bias, and antigen-driven hypotheses. For example, the antigen-driven persistence model is not directly tested, so that suggesting future multi-center studies to validate across diverse cohorts is warranted. To address antigen-driven persistence limitations (Page 15, Lines 8-12), functional validation studies to confirm antigen-driven B-cell expansion should be considered. Furthermore, peptide screening assays to identify putative antigens involved in pcMZL would be of great interest, though might be beyond the scope of this study. Also, longitudinal studies tracking B-cell evolution in pcMZL patients over time might be considered as future directions (Page 24, Conclusion).

As suggested, we added a direct comparison of two longitudinal samples from a patient with rB-LP and pcDLBCL-LT respectively (new Supplementary Figure 6). These showed that within recurring lesions within 6 and 12 months respectively the level of clonal expansion and phenotypic composition did not change.

Nevertheless, we also fully agree that our study can only be a starting point for larger studies that validate our findings. We believe that we provide the strongest evidence yet that indolent CBCL may be primarily antigen-driven reactions. Therefore, our data provides a good starting point for such in-depth and larger scale studies. We highlighted this in our discussion:

[...] Subsequent studies are needed to functionally validate this finding and identify these putatively driving antigens. [...]

Reviewer #2 (Remarks to the Author):

This paper tackles the interesting challenge that we have in hematopathology on how to distinguish reactive lymphoid lesions harboring clonal populations from truly neoplastic lesion, and when one should use the term "lymphoma." There are several lymphoproliferative disorders that have clonal B cell populations, but are clearly not malignant, and we, as a field struggle with how to deal with these. This is certainly an important question and one of clinical impact.

Griss et al ambitiously tackle this problem using single cell sequencing of several of these relatively rare lesions, and report some interesting findings and make some valid arguments about whether particular lesions may be reactive or neoplastic on the basis of the dominance of the main clone, whether the lesion contains "aberrant cells" and whether there was persistence of germinal center reaction. The authors illustrate that our routine approaches in pathology to distinguish lymphoma or not are quite flawed. This was best illustrated by the two examples of the switched and non-switched pcMZL where the authors demonstrate that this criteria is not sufficient to distinguish a lymphoma from a reactive process.

While the manuscript and the data in it were quite interesting, the findings remain mostly descriptive, and are not sufficient to make the broad definitive conclusions that the authors make about these disease entities. There are a number of weaknesses along these lines:

1. Given the amount of heterogeneity present in these lesions, the number of samples analyzed for each entity is very small. In addition, the authors do not describe how tumors are sampled, and there is no consideration regarding sampling variation, which would impact the % of a clonal population present and the % of supportive cells present, for example. It would be nice to see a few examples of multiple sampling from one tumor to assess the level of variability that may be present just in sampling different areas.

For the revision of the manuscript we were able to acquire two independent samples from a patient with newly diagnosed rB-LP which enabled us to assess the spatial heterogeneity (new Supplementary Figure 5). Additionally, we added an in-depth analysis of matched samples from a patient with rB-LP and pcDLBCL-LT which were acquired 6 and 12 months apart respectively (new Supplementary Figure 6). In all of these cases, the level of clonal expansion and phenotypic composition remained highly similar. Finally, we more than doubled our IHC cohort to a total of 40 samples, which further confirmed our scRNA-seq based characterisations (new Figure 2). Thereby, we believe that we were able to demonstrate the robustness of our method, the consistency of the lesions in space and time, as well as the consistent results among a large patient cohort.

2. These are challenging entities to diagnosis and nomenclature has changed over time. There is no description of how the diagnoses were verified, pathology review, and validation on how the patients were clinically staged. There is limited clinical data, survival data, or consideration of age/gender, etc. This is critical information needed if one will make definitive statements about particular entities.

We apologize that this information was inadvertently omitted in our methods section. All histopathological samples were reviewed by two independent histopathologists. Furthermore, all patients with CBCL were staged using full body CT scans to rule out systemic disease. We adapted the methods section accordingly:

Histopathological diagnoses were confirmed by two independent board-certified histopathologists. All patients with pcMZL, pcFCL, and pcDLBCL-LT were further staged using full body CT scans to rule out systemic disease.

Age, gender, and disease stage are all available in Supplementary Table 1. Yet, the cohort size is insufficient to perform sex- or age-based analyses within each entity. Nevertheless, we added additional figures to showcase the clonal distribution in each individual sample (new Figure 3D) which highlights the similarity of all samples within a disease group.

3. I think that the way different B cell populations are defined are limited and use minimal markers. For example, aberrant populations seem to be mostly defined by having variation from the normal expression of 2-3 markers.

As suggested we extended the characterisation of all presented phenotypes. For all analyses, we furthermore added supplementary tables that showcase the complete marker profiles of each individual B cell subtype.

The text of the main manuscript as extended to highlight the key findings:

We subsequently performed a differential expression analysis of all of these subtypes (Supplementary Table 2 - diff genes CBCL B cells). Matching the phenotypic assignment, naive B cells primarily showed an up-regulation of activation and signalling associated genes, such as CD69, CD83, and CD44, as well as CXCR4 which is important for chemotaxis. Moreover, they expressed FCER2 (CD21) linked to B cell maturation. Next to immunoglobulin associated genes, genes highly expressed in plasma cells were mainly associated with protein synthesis, such as DNAJB9, DNAJC3, PDIA4, and secretion, such as SEC61A1, SEC24D, and SAR1B. Germinal center cells showed a strong expression of cell cycle regulating genes, such as PLK1, CDC20, CCNA2, CCNB1, CCNB2, CDK1, and CDCA3. Finally, the group of aberrant B cells uniquely overexpressed energy metabolism associated genes, such as member of the electron transport chain for APT production (NDUFA4, NDUFB11, NDUFB4, NDUFB7, NDUFB10, NDUFB9, NDUFV2, NDUFA11, NDUFS6, NDUFS5), but also oxidative phosphorylation (COX5A, COX7B, and COX6A1) and glycolysis (GAPDH, LDHA, LDHB, TPI1, ENO1, PGAM1). This clearly shows that these aberrant B cells were metabolically highly active, matching their proposed malignant phenotype.

4. Independent validation of expression data was quite limited

In order to improve the validation of our scRNA-seq based characterisation we more than doubled our IHC based cohort to a total of 40 samples. As an example, this also contained a matched sample from one of our pcDLBCL-LT patients. There, IHC analyses confirmed the presence of a large CD27+ B cell population as suggested by the scRNA-seq data. We therefore believe that we present a comprehensive protein-based validation of our key findings.

5. Assessment of data QC seemed quite limited

We performed stringent Q/C validation for the scRNA-seq data. These included filtering based on the percentage of mitochondrial RNA, as well as the number of genes / reads per cell, following current best practices.

We extended Supplementary Data 2 to include all of this information on a sample level, together with detailed information about the BCR coverage which is key to identify the top expanded clone.

During the analysis of the spatial heterogeneity we inadvertently evaluated the effect of different sample quality (new Supplementary Figure 5). Due to technical and biological variation the number of recovered cells differed considerably between these two simultaneously collected samples. Nevertheless, both the amount of clonal expansion, nor the assessment of the phenotypic composition was affected by these differences.

We therefore hope that we can transparently show that our data is of consistently high quality to support our claims.

For all of these reasons, the data, while interesting are not, in my opinion, sufficient to support the broad claims made about pcMZL, pcFL, etc.

We therefore hope that we were able to sufficiently validate our data to support our claims.

Reviewer #3 (Remarks to the Author):

The authors have generated and analyzed one of the largest single cell RNA-Seq datasets generated for cutaneous B cell lymphomas. These include pcMZL n=9, pcFCL n=5, pcDLBCL-LT n=4, rB-LP n=5. They do standard informatic analyses to cluster the cell types, to identify the putative clone, and to elucidate the putative cell of origin.

The field of cutaneous B cell lymphomas is a burgeoning field with new insights that can have broad implications for 1) cutaneous lymphomas, 2) cutaneous immunology, and 3) B cell biology. They share some markers about the phenotypic diversity in pcMZL that contrasts with the more uniform distribution of the cells in the other cutaneous lymphomas.

Major points:

The conclusions from this dataset are limited, which I will discuss here.

1. One of the major questions in the field is how cutaneous and systemic lymphomas are different? Why are the cutaneous B cell lymphomas so indolent? The description of the differences between the systemic and cutaneous lymphomas is inconsistent. A fundamental feature of cutaneous lymphomas is that they are genetically and proteomically distinct from the systemic lymphomas. How much of this can be attributed to the differences in the cell of origin? Restricting much of the analyses to skin lymphomas and skin controls limits the impact. Are the pcDLBCL-LTs a different cluster because the appropriate cells of origin, ABC-type cutaneous lymphomas, are not represented in this dataset? or because they are truly different?

We very much thank the reviewer for this comment! As suggested we extended our manuscript with a completely new comprehensive analysis and comparison between the cutaneous and systemic

lymphomas (new Figure 3). There, we found that both pcDLBCL-LT and sDLBCL share unique features that differentiate them from all other entities. Nevertheless, we also found unique features for each entity that explains some of their differences. With respect to the cell of origin our data does support the notion that pcDLBCL-LT originates from antigen experienced, activated B cells. This overall supports the similar naming of the types of DLBCL, while there are also subtype specific features.

Similarly, we are now able to show more conclusively that gastric MALT lymphomas have a unique phenotype that is different to other lymphomas. This further highlights that pcMZL lacks any of the features associated with true lymphomas.

2. Phenotypic features that distinguish cutaneous and systemic lymphomas are almost completely ignored. Many of these are derived from genetic differences between the cutaneous and systemic counterparts. These fundamental biological insights are not addressed. For example, pcFCL have significantly less BCL2 expression than their systemic counterparts. These comparisons are required to make the analyses interesting.

As part of our new in-depth comparison of cutaneous vs. systemic lymphomas (new Figure 3) we also performed differential expression analyses. These also highlighted the already known higher expression of BCL6 in pcFCL and the expression of BCL2 in sFCL. Additionally, we found a large number of additional differences that may be useful for further validation.

The complete results of these differential expression analyses were additionally reported in the new Supplementary Data 4.

3. The use of cutaneous controls is unusual. While there are potential lymphoid aggregates in the skin; traditionally, many of these germinal zone aggregates are formed in the lymph node. It would be better to compare to see if the cells here are similar to the ones found in lymph nodes or different. If so, how?

We fully agree that normal healthy skin cannot be used as a control for a “normal” B cell infiltrate. Nevertheless, we also believe that some kind of baseline is needed, as B cells physiologically do not infiltrate the skin. We therefore used rB-LP samples as primary control.

Nevertheless, as suggested we now added additional data from reactive lymph nodes to our analysis (diagnosed as “reactive hyperplasia” by a pathologist) (new Figure 3) that serve as another control, better suited to represent physiological B cell driven reactions.

4. There is almost no discussion about the microenvironment re: signals between pro-tumor and anti-tumor cells with the lymphoma cells. A wealth of data suggest that many mutations in these CBCLs occur in cell surface proteins that are predicted to mediate interactions with the environment, such as TNFRSF14, FAS, and/or MHC class I. These are not addressed.

As suggested, we performed an in-depth interaction analysis using CellChat. This highlighted key differences between indolent and aggressive CBCL which are now presented in detail in the new Figure 4A. Overall, we believe that the data supports the notion that indolent CBCL are primarily antigen-driven inflammatory reactions.

5. These tumors are rare, meaning that it is difficult to expand the dataset to ensure generalizability. In that setting, the manuscript will require orthogonal validation through spatial and/or surface IHC to describe how these markers can be leveraged across larger datasets.

Unfortunately, scRNA-seq is the only method that can combine phenotypic characterisation with clonal assignment. Additionally, especially in pcMZL and pcFCL, the clonally expanded B cells phenotypically do not differ from the significant polyclonal B cell bystander infiltrate.

In order to nevertheless validate our phenotypic assignments, we more than doubled our IHC cohort to a total of 40 samples. We also highlight a specific pcDLCBL-LT case where we have matching scRNA-seq and IHC data. scRNA-seq suggested a comparably large population of CD27+ B cells, which was also seen in the IHC data. Therefore, we believe that we present good evidence that our scRNA-seq data is representative of the underlying biology.

Minor.

1. The manuscript is written in a way that makes it difficult to read for non-experts in the field. Tables that describe the differences between clinical entities and their systemic counterparts would make it easier to read.

As suggested, we added graphical summaries of the B cell development together with the distribution of clonally expanded B cells to simplify the data presentation (new Figure 3A).

2. Genetic information can be gleaned from well curated datasets. Are any mutations observed in these tumors?

Previous studies were unable to find consistent mutations in CBCL and these are not routinely assessed. Our study was not designed to assess the genetic background of these diseases, which would have required a different setup. A search for such driver mutations would have been outside the scope of our study.

3. The choice of controls such as MALT lymphomas are hard to understand compared to other systemic lymphomas.

We chose gastric MALT lymphoma as it is the most frequent non-Hodgkin marginal zone lymphoma in humans. Also, pcMZL have previously been termed “cutaneous MALT lymphoma”, assuming that pcMZL are the cutaneous counterpart of gastric MALT lymphomas. We therefore felt that it was an appropriate comparison for cutaneous pcMZL. Nevertheless, we extended our control samples to reactive lymph node tissue as an additional control for physiological B cell reactions (new Figure 3).

4. Description of the "bystander" B cells would be useful. Are these cells defined by merely the absence of the clonal BCR? if so, could this be a technical error? do they express alternative BCRs?

We adapted the manuscript to highlight that these are polyclonal B cells:

In each sample, we were thus able to unambiguously distinguish the clonally expanded (presumed malignant) B cells from the polyclonal bystander infiltrate (Supplementary Data 2).

We further extended Supplementary Data 2 to also include the number of BCR negative B cells, which were excluded from any comparative analyses. This was additionally highlighted in the new Supplementary Figure 4 where we show that missing BCR data is unlikely to affect the interpretation of our pcMZL-related conclusions.

Reviewer 1

Thank you for including the helpful comparison between cutaneous and systemic lymphomas, as well as context on the challenges with paraffin-based BCR sequencing. I recommend adding a brief acknowledgment of the limitations of current diagnostics and how your additional work aims to address them, even if beyond the scope of this study.

Response: We very much thank the reviewer for this comment. Our own evaluation of new methods is still too early to share them in more detail. Nevertheless, we adapted our discussion to highlight that scRNA-seq can be used as a reference to evaluate new methods:

... Our data thus holds the promise that scRNA-seq of cutaneous samples may be sufficient to distinguish between primary cutaneous and systemic subtypes of B cell lymphomas arising in the skin and can act as a reference to evaluate new methods.

Thank you for addressing this point directly. The clarification of how your findings influence clinical management and align with current classification debates adds valuable context and highlights the paradigm-shifting potential of your work.

Response: We thank the reviewer for these positive comments.

Thank you for this important addition. The discussion of diagnostic utility is compelling. I recommend briefly expanding on how scRNA-seq may aid prognostication in future applications, particularly in distinguishing clinically indolent versus aggressive disease trajectories.

Response: As suggested, we added two statements to our discussion to highlight potential future prospects, with regards to diagnostic accuracy:

... Our data thus holds the promise that scRNA-seq of cutaneous samples may be sufficient to distinguish between primary cutaneous and systemic subtypes of B cell lymphomas arising in the skin and can act as a reference to evaluate new methods.

... Nevertheless, our study is limited with respect to the sample size and demographically homogeneous cohort. Therefore, further multi-center clinical studies are needed to validate our findings, including whether the rate of clonal expansion correlates with clinical outcome.

Thank you for this clarification. Since your data do not yet allow for conclusions on therapeutic intensification in pcMZL, I suggest briefly noting this limitation in the Discussion and stating the need for future studies stratifying treatment possibly based on clonal expansion.

Response: We extended our discussion to clearly state that our results offer new hypotheses that need to be addressed in future clinical studies:

... Nevertheless, our study is limited with respect to the sample size and demographically homogeneous cohort. Therefore, further multi-center clinical studies are needed to validate our findings, including whether the rate of clonal expansion correlates with clinical outcome.

Thank you for addressing this clinically relevant point. The revised Discussion makes a compelling case for antigen-targeted strategies. If feasible, consider briefly mentioning candidate antigens or classes of antigens (e.g., Borrelia-associated or autoantigenic targets) to increase translational relevance.

Response: We adapted our discussion to highlight previous reports where both pcMZL and pcFCL were linked to infections, including Borrelia, and resolved with antibiotic therapies.

This matches multiple reports where both pcMZL and pcFCL were linked to infections⁵⁰⁻⁵² and responded to antibiotic therapies or vaccinations⁵³⁻⁵⁵.

Thank you for significantly expanding the IHC validation cohort and including longitudinal and spatial comparisons. It significantly strengthened your manuscript. For transparency, you may wish to note the need for further validation in independent, multicenter cohorts as a future direction.

Response: As mentioned above, we added a clear statement that our results need to be validated in future studies.

... Nevertheless, our study is limited with respect to the sample size and demographically homogeneous cohort. Therefore, further multi-center clinical studies are needed to validate our findings, including whether the rate of clonal expansion correlates with clinical outcome.

Thank you for including the comparison to publicly available dataset (Ramelyte et al.) These external data significantly reinforce the robustness of your conclusions and strengthens the manuscript.

Response: We thank the reviewer for these positive comments.

Thank you for implementing multiple-testing corrections and clarifying this in the Methods and Results sections.

Response: We thank the reviewer for these positive comments.

Thank you for these substantial improvements. The updated figures and extended legends enhance clarity and accessibility, particularly for non-specialist readers.

Response: We thank the reviewer for these positive comments.

Thank you for adapting the supplementary figures and improving the legends. The B-cell evolution schematic is highly informative and will help a broader readership engage with your findings.

Response: We thank the reviewer for these positive comments.

The revised manuscript provides promising evidence and outlines the need for further validation. To strengthen transparency, please consider explicitly acknowledging the limited sample size and demographic representation as part of the study's limitations.

Response: As suggested, we clearly highlight that our study has these limitations.

... Nevertheless, our study is limited with respect to sample size and a demographically homogeneous cohort. Therefore, further multi-center clinical studies are needed to validate our findings, including whether the rate of clonal expansion correlates with clinical outcome.

Reviewer 2

The study tackles the challenging area of how to distinguish indolent primary cutaneous lesions and how we should define malignant lymphomas in this setting. The authors have done this through detailed single-cell RNA sequencing analysis of these primary cutaneous lymphomas. Though the paper as originally submitted was definitely intriguing, this author identified a number of weaknesses, including the small n size for the conclusions being made, no independent validation, description of how these cases were classified histologically, limited phenotypic characterization, description QC metrics. The authors have since added new and informative samples and phenotypic and other analyses that address the concerns raised previously. The paper is strong and provides very intriguing evidence that these indolent primary lymphomas are locally driven and provide a basis for future discussion regarding how these should be classified (probably not as malignant lymphomas).

Response: We thank the reviewer for these comments and for helping to improve this manuscript. We are glad that the reviewer noted significant improvements since the last submission.

However, I have conceptual concern that I believe should be addressed:

Much of the evidence that pcMZL and pcFCL may be a locally driven process (and perhaps not truly represent malignant lymphoma) rests in the fact that pcMZL, pcFCL exhibit persistent

germinal center reaction, not observed in pcDLBCL and gastric MALT. This represents very strong evidence that pcMZL (especially with the relatively small clonal B populations seen in pcMZL) represents a localized lymphoproliferative disorder and not a true lymphoma. However, for pcFCL where you have shown larger relative clonal populations, it seems as if there is not sufficient evidence to make the same statement. I wonder about the comparison of pcFL to sFL. FL is a systemic disease and considered malignant, but it also maintains ongoing somatic hypermutation, and also harbors the important elements of the microenvironment, on which these cells are thought to depend. Unlike pcMZL, your data shows that pcFL has the second highest level of ongoing SH with a narrow range (similar to pcDLBCL). It also seems to have a higher clonal percentage. Therefore, the distinction between sFL and pcFL on the parameters that you are using to reclassify pcMZL is less clear. I think this should be addressed in the discussion by either 1) providing more evidence or further explaining the distinction between pcFCL and sFL in terms of ongoing SH, clonal populations, microenvironment - or restricting the main hypothesis about the biology of primary cutaneous lymphomas to pcMZLs, and acknowledging that pcFL may represent a neoplastic process whose anatomic restriction remains less understood. To this end, I would suggest modifying the title to focus on "pcMZL" and not generally primary cutaneous lymphomas.

Response: We fully share the reviewer's view that our data is less clear on pcFCL compared to pcMZL. We tried to highlight this in Figure 5 where we tried to clearly show that we do believe that pcFCL cells do acquire some kind of genetic alteration that causes their proliferation. Yet, in contrast to nodal FCL the support cells required for SH are physiologically not present in skin. A true lymphoma such as pcDLBCL-LT therefore lacks these support cells as it is unable to recruit them (in contrast to nodal DLBCL where these support cells are already present). We therefore feel that our hypothesis that pcFCL originates from an initial antigen-driven response is supported (yet not proven) by our data and a more likely explanation than altered B cells that are able to recruit the complete environment that provides necessary survival signals without any signs of systemic disease.

As it was not our intention to present the evidence for pcFCL at the same level as pcMZL, we adapted our discussion as follows:

... Yet, these clones were still undergoing somatic hypermutation as part of the germinal centre reaction, which nevertheless is also observed in sFCL. ...

... Thereby, the clone may be able to expand, yet still requires the support from the germinal centre environment. As this microenvironment is physiologically not present in skin, we believe that it is most likely that pcFCL develops from a previous antigen-driven reaction. This dependence locks it in its single phenotype and may explain pcFCL's indolent behaviour. Nevertheless, further studies are needed to determine whether pcFCL should be classified as lymphoma or lymphoproliferative reaction.

Overall, this is an excellent thought-provoking manuscript that provides new data to help us better understand these primary cutaneous lesions.

Response: We thank the reviewer for these positive and supporting comments, and for the abovementioned insights, which for sure has helped to significantly improve our paper.

Reviewer 3

The authors present a minimally revised manuscript. My concerns have not been adequately addressed. I will again list them here. In sum, the inability to attempt these analyses suggest a broad ignorance of the field, the underlying B cell lymphoma biology, the techniques accessible to them, and the statistical rigor that additional analyses would provide. The points outlined below are not intended to be complete but rather to highlight the overconfidence by which the authors tried to address scientific points raised by myself and the other reviewers.

1. The fields of B cell biology already capture parallels between normal B cell biology and B cell lymphomas. This observation does not make their paper impactful. In fact, these descriptions and analogies are baked into their disease entities. The follicular, marginal zone, and diffuse B cell lymphomas have many clear analogies to B cells in different phases of differentiation and expansion. These cells of origin are, in fact, even more detailed as the diffuse large B cell lymphomas comprise multiple clinically relevant phenotypic differences between activated B cell type and germinal cell B cell type. Moreover, there are additional differences amongst the ABC-type leading to clinically relevant differences between nodal and extra nodal where these can be broken down to systemic ABC-type DLBCL, CNS/test ABC-type DLBCL+/- skin, and mediastinal DLBCL. These have now been distributed across additional subtypes based on genetics and transcriptional programs. They have clear differences that can be identified across disease subtypes and across cells-of-origin. I and the other reviewers had suggested a careful and thoughtful consideration of these differences to highlight what is truly novel biology unlocked by their single cell RNA-Sequence analyses. These are inadequately covered by the new Figure 3. This should have been thoughtfully done with known markers across disease subtypes and in fact belies an insufficient understanding of the field.

Response: We thank the reviewer for pointing out these important factors associated with lymphoma biology, namely genetic mutations and copy number variation (CNV). We are well aware of the fact that numerous single-cell RNA sequencing papers exist extrapolating genetic differences from their transcriptomic dataset.

However, we deliberately chose the droplet-based 10x platform due to the unprecedented number of cells that can be analysed. We do believe that only this very large number of cells compiled in our dataset enabled us to arrive at the presented insights. Nevertheless, this technology comes with specific drawbacks in comparison to full-transcript platforms such as SmartSeq 2 / 3. Droplet-based data is characterized by shallow per-cell coverage, high sparsity, and significant dropout rates, which fundamentally limit the sensitivity and specificity of CNV and point mutation detection.

Tools to detect point mutations, such as RESA (Zhang *et al.*, [10.1186/s13073-023-01269-1](https://doi.org/10.1186/s13073-023-01269-1)) are therefore explicitly not recommended to be used with 10x-based data. Similarly, SComatic (Muyas, *et al.*, [Nat Biotech](https://doi.org/10.1038/s41587-022-01269-1)) requires at least five reads of the same region and cell type to call a mutation, which is only reached for very small portions of the genome in 10x-based data.

In the beginning of our data analysis we did use Numbat to derive CNV data from our samples. We then used our BCR data as gold standard to assess the accuracy of Numbat's CNV-based clonal assignment. Matching published data, Numbat was only able to identify a clone in a subset of the samples, while it overestimated the clonal expansion in others (Figure 1).

Figure 1: Number of cells as part of the top clone per sample and disease as estimated through BCR-sequencing and CNV-calling (Numbat)

Samples, where Numbat identified considerably higher levels of clonal expansion included non-B cells in this assignment (Figure 2).

Figure 2: CNV and BCR-based clonal assignment in sample 146 (MZL). Numbat identified 2 clones that spanned both B and T cells. The Numbat-based clones did not overlap with the top-expanded clone identified through BCR sequencing but contained only B cells identified as polyclonal.

Simultaneously, Numbat was not able to identify a clone in samples with considerable clonal expansion based on BCR sequencing (Figure 3).

Figure 3: CNV and BCR-based clonal assignment in sample 200 (gastric MALT). Numbat was not able to identify any clonal expansion, while BCR-sequencing found virtually all memory B cells to be part of the top-expanded clone.

Overall, this analysis clearly shows that this method is unsuited to derive reliable CNV-based results from our data, matching known limitations of these methods in 10x-based scRNA-seq data.

2. The differences from tumor to tumor and across subtypes strongly correlate with genetics, suggesting a determinative relationship between mutations and phenotypes. Cancers are subject to both copy number and point mutations. There are known mechanisms to identify putative copy number mutations and/or point mutations across single cell RNA-seq datasets. While the authors may not have initially intended to analyze these, their disregard for this suggestion suggests that they under appreciate the importance of genetics to lymphoma biology, which makes it difficult to appreciate whether they understand the field at all and/or the bioinformatics tools that they are deploying.

Response: As mentioned above, our droplet-based data is unsuited to characterise point mutations and CNV-analyses proved to be too unreliable in 10x-based data.

For example, again, there are canonical DLBCL mutations in systemic lymphomas, in skin lymphomas, and shared across the two. There are similar mutations in the other cutaneous B cell lymphoma subtypes.

Response: While we are aware of the published mutational landscape of nodal / systemic and primary cutaneous lymphomas, it was not our intent to replicate these studies. As mentioned above we are unable to reliably derive this information from our data.

3. I have not reviewed or read a lymphoma manuscript with single cell RNA-Seq that has not described the host response to the tumor. CellChat does not describe the tumor microenvironments. It does not describe the cellular composition of the lymphomas across patients and within individual patients.

Response: We thank the reviewer for raising these concerns. During the last round of reviews for this manuscript, Reviewer 3 stated that “There is almost no discussion about the microenvironment re: signals between pro-tumor and anti-tumor cells with the lymphoma cells. A wealth of data suggest that many mutations in these CBCLs occur in cell surface proteins that are predicted to mediate interactions with the environment, such as TNFRSF14, FAS, and/or MHC class I. These are not addressed.” As the reviewer was specifically asking about interaction between cells, we used CellChat, one of the most frequently used methods to analyse interactions in scRNA-seq data. Moreover, we present the overall composition of the tumour microenvironment in Supplementary Figure 1B.

The tumors themselves have genetically-encoded differences in immune ligands. these are not addressed in a systematic way. Their de-prioritization based on CellChat does not mean that they have been actively down regulated by the tumor and/or if this has effects on other cells in the microenvironment. The decision not to pursue this suggests an inadequate appreciation for what tumor immunology represents.

Response: We entirely agree with the reviewer that genetically encoded differences exist for immune ligands in cancers. As shown in several methods (Muyas, *et al.*, Nat Biotech), 10x-based scRNAseq is an inadequate method to assess point mutations. We also do not understand the criticism of CellChat or any receptor/ligand interaction platform to assess the tumor cell interaction with the microenvironment.

4. The generalizability of these findings. These studies are the smallest genomic-clinical cohorts that I have seen. The studies with bulk RNA or DNA-Seq have much higher samples which enable broader generalizations of their data. Since these are rare tumors, we had suggested spatial transcriptomics to confirm orthogonally these findings, identify novel spatial interactions, and enable extension of their work to new samples for which fresh single cells are not available. Their claims that this would limit the ability to track antigen-responsive cells (Especially for matched tumors) reflects an ignorance about the potential of integrating these datasets. Again, this highlights an overconfidence in their ability to make conclusions based on their limited understanding of the lymphoma biology, the technical approaches, and the statistical rigor.

Response: We were initially excited to hear from Reviewer 3 that “The authors have generated and analyzed one of the largest single cell RNA-Seq datasets generated for cutaneous B cell lymphomas,” particularly as we are integrating more than 250,000 cells, an unprecedented number in the field of cutaneous B-cell lymphomas to date. We are thus not entirely sure how to approach the criticism raised in this paragraph about us presenting the smallest group. Again, we politely emphasize that we are not providing a genomic-clinical cohort here. scRNAseq is a transcriptomic approach that, as we feel, does not adequately address genomic differences. We certainly agree that there are larger bulk RNA or DNA sequencing studies out there in this field, but are unsure how this relates to our single-cell dataset.

We want to re-emphasize that there are currently no spatial transcriptomic platforms available that can combine BCR information with spatial information due to technical limitations, both with

truly transcriptomic assays as well as probe-based assays. It would of course be desirable to detect specific BCR sequences in spatial transcriptomic datasets, and we hope that this technology will become available in the future.

Reviewer 2

We very much want to thank the reviewer for the thorough feedback on our manuscript! Overall, we believe that our view relating to pcFCL is well aligned with the reviewer's comments. However, we acknowledge that certain passages may have been unclear and we have revised these sections to improve clarity. As key changes we:

- Adapted the abstract to clearly state that we do not propose a re-classification of pcFCL but do observe features of a true lymphoma.
- Re-wrote the respective paragraphs of the discussion to highlight that the known driver mutations in pcFCL very well match the rate of clonal expansion we observe.

With respect to the origin of pcFCL, we do feel that our interpretation of the data is in-line with the current consensus in the literature. We therefore added an additional reference to Yu *et al.*, where the development of TLS within lesions of Hidradenitis suppurativa is also interpreted as proof of an antigen-directed immune reaction. Please see our point-by-point responses below:

As mentioned previously, the data in this paper are interesting and support the idea that pcMZL may be a localized antigen-driven lymphoproliferative disorder as opposed to a true lymphoma. However the evidence in the manuscript that the same may be claimed for pcFL is much weaker in comparison. My suggestion was either to provide more evidence supporting this claim for pcFL – or restricting the claim for primacy cutaneous MZL both I the discussion in the title.

The authors' response is insufficient and presents the data in the manuscript and hypotheses based on this data - in a vacuum , totally ignoring what is known in the field about cutaneous and systemic lymphomas:

"In contrast to nodal FCL the support cells required for SH are physiologically not present in skin".

- It is actually well known that many of the microenvironmental cells relevant to pCFL are normally present in the skin, though with different frequency and organization, including CD4+ T cells, CD8+ cytotoxic T cells, dermal dendritic cells, macrophages and stromal cells. While Tfh cells are rare in skin, they can be recruited during inflammation.

Response: We very much thank the reviewer for this detailed assessment! As suggested, we additionally adapted the abstract to clearly state that we do not propose a re-classification of pcFCL.

Our data thus indicate that pcMZL and pcFCL, similar to rB-LP may be driven by (a yet unknown) antigen. While our data indicates that pcFCL exhibits some features of true lymphomas, it clearly supports the classification of pcMZL as a lymphoproliferative disease.

However, we believe our findings provide strong evidence that pcFCL originates from an antigen-driven response as it is associated with a functional germinal center reaction - as recognized by the reviewer. Regarding the reviewer's comment on TLS in the skin, we would like to emphasize that their formation is rare and has only been described in the context of anti-tumor immunity and very few inflammatory skin diseases, such as Hidradenitis suppurativa. Many research groups now consider this as evidence supporting the classification of Hidradenitis suppurativa as an autoimmune disease (Yu et al., Immunity, 2024, 10.1016/j.immuni.2024.11.010). We therefore strongly believe that this context warrants different considerations than those applied to lymph nodes.

"A true lymphoma such as pcDLBCL-LT therefore lacks these support cells as it is unable to recruit them (in contrast to nodal DLBCL where these support cells are already present)."

- This statement does not make sense. Both pcDLBCL-LT and nodal DLBCL are characterized by sheets of large neoplastic and are independent of microenvironment support cells (for the most part). Both PC DLBCL-LT and nodal DLBCL are both malignant neoplasms due to their genetic alterations and obviously does not need to recruit support cells. The differences between these lymphomas do not lie in difference in ability to recruit support cells, but major differences in genetic alterations specific for each entity. These genetic differences are very well described.

Response: We apologize for the misunderstanding and believe that our presented data perfectly matches the reviewer's comment. We now clearly highlight that these genetic differences are known and were already described:

This further aligns with previous studies that were unable to find consistent driver mutations in pcMZL¹⁷ in contrast to pcDLCL-LT⁵⁸.

"We therefore feel that our hypothesis that pcFCL originates from an initial antigen-driven response is supported (yet not proven) by our data and a more likely explanation than altered B cells that are able to recruit the complete environment that provides necessary survival signals without any signs of systemic disease."

-- This statement is also problematic.

1) It is still not clear what component of the current data supports the idea that pcFCL originates from an initial antigen-driven response. That data is lacking.

Response: As we highlight in our discussion, there is consensus in the literature that TLS only form as part of an antigen-driven reaction. We are further convinced that our data clearly shows that pcFCL is showing a functional germinal center reaction - which only takes place in TLS outside of lymph nodes. In clonally expanded B cells we could clearly track the somatic hypermutation and identify all required support cells. We further support this statement with a new reference to a study in Hidradenitis suppurativa that argues that due to the presence of

TLS, HS should be regarded as an auto-immune disease as the identified antigens targeted by the B cells were found in the epidermis.

Extra nodal germinal centre reactions are generally considered a sign of the formation of tertiary lymphoid structures⁴⁶, are primarily observed in cancer⁴⁷ and autoimmune diseases⁴⁸ and are a sign of chronic, antigen-specific immune responses⁴⁹, especially in the skin⁵⁰.

2) Stating that an entity originates from an initial antigen-driven response is very different than saying an entity is an antigen-driven lymphoproliferative process that is not lymphoma. Gastric MALT - which you correctly describe in the manuscript as true lymphoma originates from an antigen-driven response – namely *H.pylori*. Sometimes, these lymphomas remain responsive the *H.Pylori* eradication, and sometimes they become independent of *H.pylori* stimulation as a result genetic alterations. So whether pcFL initially originates from an antigen-driven response is sort of irrelevant to whether it should be called a lymphoma.

Response: We apologize for this misunderstanding and further clarified, that we are not proposing that pcFCL should not be considered a true lymphoma. We therefore adapted the abstract accordingly (see above). Further, our discussion clearly states that our data cannot answer this question:

Nevertheless, further studies are needed to determine whether pcFCL should be classified as lymphoma or lymphoproliferative reaction.

With respect to the relevance, we do believe that this is of potential clinical relevance. As pointed out by the reviewer, antibiotic *H. pylori* eradication is an established therapeutic approach for gastric MALT lymphoma. Our data supports previous case reports that this may be effective in a subset of pcFCL cases.

This matches multiple reports where both pcMZL and pcFCL were linked to infections⁵¹⁻⁵³ and responded to antibiotic therapies or vaccinations⁵⁴⁻⁵⁶.

3) Unfortunately, the authors present their work in a vacuum.
- There are two current classifications in lymphoma – WHO and ICC (International Consensus Classification). Although the WHO5 continues to classify pcMZL as a lymphoma, the 2022 ICC has already been downgraded it to a persistent antigen-driven “lymphoproliferative disorder” based on current literature, clinical outcomes, association with *Borrelia burgdorferi*, – so this concept is not new. This should be mentioned in the manuscript. The current data mainly serves to support this point.

Response: We are very well aware of this current inconsistency in international classifications and discuss this point in detail in the second paragraph of our manuscript’s introduction and the first sentence of the discussion. Throughout the manuscript, we further clearly state that our data supports the existing concept that pcMZL may represent a lymphoproliferative disorder, f.e.

We therefore, for the first time, present molecular evidence that aligns with the concept that pcMZL is a lymphoproliferative disorder and not a true lymphoma.

- There is some suggestion in the literature that pcFL also originates from an antigen-driven response, but no great definitive evidence – neither is there in this manuscript. However, there is clear evidence that pcFL exhibits some of the features of a lymphoma – including for example its tendency to recur locally after excision, and predominance of clonal cells within the lesions. That being said, it's not as aggressive as pcDLBCL because it tends to stay in the skin – even when it recurs, and prognosis is excellent, while pcDLBCL has poor prognosis.

Response: We very much thank the reviewer for this comment. We feel that our data perfectly matches these observations and unites these concepts. We therefore re-wrote the respective paragraphs in the discussion to clarify this point (see below).

- There are papers that demonstrate the phenotypic (BCL2 expression) and genetic alterations (CBP, MLL2, KTM2D, etc) mutations that underlie differences between pcFL and systemic FL that infiltrate skin – this should not be ignored in the manuscript as part of the discussion!!

Response: We very much thank the reviewer for this comment and re-wrote the respective paragraph in the discussion to clearly show that specific driver mutations are known in pcFCL.

"As it was not our intention to present the evidence for pcFCL at the same level as pcMZL, we adapted our discussion as follows: ... Yet, these clones were still undergoing somatic hypermutation as part of the germinal centre reaction, which nevertheless is also observed in sFCL. ... Thereby, the clone may be able to expand, yet still requires the support from the germinal centre environment. As this microenvironment is physiologically not present in skin, we believe that it is most likely that pcFCL develops from a previous antigen-driven reaction. This dependence locks it in its single phenotype and may explain pcFCL's indolent behaviour. Nevertheless, further studies are needed to determine whether pcFCL should be classified as lymphoma or lymphoproliferative reaction."

To this reviewer, this explanation is unclear (I am not sure what author is trying to communicate) and ignores much of what is already in the literature on this topic, as described above

- PMID: 40227781
- PMID: 33045225
- PMID: 33560380

I would advise authors to really integrate what has already been published in the field, and to focus your conclusions on pcMZL as being not lymphoma - and not pcFL.

Response: We very much thank the reviewer for highlighting these publications! While we already reference the study by Zhou *et al.* to highlight the mutational landscape in pcFCL, we also added a reference to the study by Barasch *et al.* Furthermore, we adapted the respective

paragraph to improve readability and highlight the fact that we do not propose a re-classification of pcFCL:

In contrast to pcMZL, pcFCL exhibits a comparably high level of clonal expansion. This matches previous studies describing putative driver mutations in pcFCL^{59,60}. These mutations likely arise during the ongoing germinal center reaction, which is known to increase the risk of such genetic alterations⁶¹. Our data further revealed ongoing somatic hypermutation - a hallmark of the germinal center reaction - also observed in systemic FCL (sFCL)⁶². This contrasts earlier reports where somatic hypermutation was not detected in a series of pcFCL cases⁶³, raising questions about whether this discrepancy stems from the higher sensitivity of our scRNA-seq approach or the existence of distinct pcFCL subsets.

Unlike sFCL, our pcFCL samples did not show signs of further differentiation but remained confined to a single phenotype. We hypothesize that while established driver mutations enable clonal expansion, sustained growth still depends on the germinal center microenvironment. Given that this microenvironment is physiologically absent in the skin, it is plausible that pcFCL originates from a prior antigen-driven reaction. This dependence may constrain the clone to a single phenotype and contribute to the indolent behavior of pcFCL. However, further studies are needed to fully elucidate these mechanisms.

Manuscript number: NCOMMS-24-82894A

Dear Editors,

Thank you very much for your invitation to resubmit our manuscript. We would like to thank all the reviewers for their comments, which we believe have significantly helped to improve the manuscript. Please find below our point-by-point responses to those comments, and we hope that these additions and corrections to the manuscript now warrant publication in your journal.

Reviewer #1 (Remarks to the Author):

- 1. This study presents a highly detailed single-cell RNA sequencing (scRNA-seq) analysis of primary cutaneous B-cell lymphomas (CBCLs), specifically pcMZL, pcFCL, and pcDLBCL-LT. It highlights new insights into their clonal evolution, transcriptomic profiles, and B-cell receptor (BCR) characteristics. The major claim that pcMZL behaves more like a lymphoproliferative disorder rather than a true lymphoma is potentially paradigm-shifting and offers novel perspectives on diagnostics, classification, and therapeutic considerations.**

Key strengths that justify publication if revised appropriately:

Novelty & Importance – This study fills a gap in molecular profiling of CBCLs, a rare lymphoma subtype.

Methodological Rigor – The use of scRNA-seq, BCR sequencing, and multiplex immunohistochemistry (IHC) provides a comprehensive multi-modal analysis. Well-Structured & Logical Flow – The manuscript is well-organized, with a clear hypothesis and stepwise data presentation.

Potential Clinical Impact – The reclassification of pcMZL as a lymphoproliferative disorder could influence future lymphoma classification and management.

The study provides strong transcriptomic insights but lacks a direct comparison to existing diagnostic standards. The clinical consequences of pcMZL reclassification need to be clarified.

Provide a comparison of scRNA-seq to current diagnostics (Page 3, Lines 20-25): compare scRNA-seq vs. IgH rearrangement PCR vs. IHC vs. flow cytometry in terms of diagnostic accuracy; include specificity and sensitivity values where possible.

Response: We very much thank the reviewer for these suggestions!

As suggested, we added an in-depth comparison between cutaneous and systemic lymphoma. These also showed the currently used difference in BCL6 and BCL2 expression between pcFCL and sFCL.

We additionally did try to perform paraffin-based BCR sequencing using Thermo Fischer's IonTorrent system. This is the diagnostic gold-standard in our hospital. Yet, out of the 12 processed samples, despite sufficient DNA, 8 failed to unclear technical issues. Out of the 4, two were deemed as polyclonal by the IonTorrent system, while the scRNA-seq data showed a clonal expansion of 50%. Based on these results the Department of Molecular Pathology now stopped the use of the IonTorrent system for paraffin-based analyses and is working with the manufacturer to analyse these issues.

In contrast to cutaneous T cell lymphoma, there is currently no gold standard to diagnose CBCL using flow cytometry.

Finally, our scRNA-seq sample set is too small to perform robust sensitivity and specificity analyses. We are currently working on different clinically usable assays that could be used on larger sample sets and subsequently be fit for routine clinical use. Nevertheless, these efforts are beyond the scope of this study.

- Thank you for including the helpful comparison between cutaneous and systemic lymphomas, as well as context on the challenges with paraffin-based BCR sequencing. I recommend adding a brief acknowledgment of the limitations of current diagnostics and how your additional work aims to address them, even if beyond the scope of this study.

Please, discuss how scRNA-seq findings align with WHO and ICC lymphoma classification criteria and clarify how findings will impact patient management (Page 16, Lines 10-15):

2. Would patients previously diagnosed with pcMZL now be managed differently?

As a first consequence patients with non-class switched pcMZL are no longer staged using CT scans at our hospital. We hope that our data will help to settle the current discussion on whether class-switched cases of pcMZL are a true lymphoma or not.

Furthermore, we very much hope and believe that our data will nevertheless spark a wider discussion into which type of treatments are warranted for the indolent types of CBCL.

Accordingly, we very strengthened these points in the discussion as follows:

We therefore, for the first time, present molecular evidence that aligns with the concept that pcMZL is a lymphoproliferative disorder and not a true lymphoma.

- Thank you for addressing this point directly. The clarification of how your findings influence clinical management and align with current classification debates adds valuable context and highlights the paradigm-shifting potential of your work.

3. Should routine scRNA-seq be considered for diagnosis or prognosis?

We believe that we were able to show that scRNA-seq can unambiguously differentiate between these entities. It would therefore definitely be advantageous to at least use such kind of data as a gold standard for studies to circumvent current diagnostic challenges.

We highlighted this in our discussion as follows:

Our data thus holds the promise that scRNA-seq of cutaneous samples may be sufficient to distinguish between primary cutaneous and systemic subtypes of B cell lymphomas arising in the skin.

- Thank you for this important addition. The discussion of diagnostic utility is compelling. I recommend briefly expanding on how scRNA-seq may aid prognostication in future applications, particularly in distinguishing clinically indolent versus aggressive disease trajectories.

4. Can you make therapeutic implications (Page 21, Lines 5-10) by discussing whether pcMZL patients with high clonal expansion might require targeted therapies or more aggressive treatment?

We are yet unable to make this statement based on our cohort as we did not have any patient with pcMZL that showed a high level of clonal expansion. Nevertheless, we strongly believe that we need to characterise patients with non-class switched pcMZL more thoroughly. Our data shows that the class-switch alone does not sufficiently identify patients with aggressive diseases.

In our dataset, only samples from pcDLBCL-LT showed a clonal expansion similar to systemic lymphomas. These patients are already receiving aggressive therapies.

We nevertheless do believe that our data offers an avenue for new treatment approaches, as highlighted below.

- Thank you for this clarification. Since your data do not yet allow for conclusions on therapeutic intensification in pcMZL, I suggest briefly noting this limitation in the Discussion and stating the need for future studies stratifying treatment possibly based on clonal expansion.

5. It would be interesting to explore potential immunotherapy targets identified from transcriptomic data. Showing clinical applicability will be beneficial and will make findings more translational and relevant to Nature Communications readers.

Our data fully supports the current notion that indolent CBCL do not require aggressive treatment. We believe that our data primarily suggest that treatment strategies should focus on identifying the

underlying antigen and potentially eliminate it to interrupt the ongoing germinal center reaction.

We explicitly added this to our discussion with a new paragraph:

Current treatment strategies for indolent CBCL are primarily targeting the B cells directly. The fact that our data strongly suggests that these are antigen-driven reactions offers an avenue for new therapeutic approaches. Subsequent studies are needed to functionally validate this finding and identify these putatively driving antigens. This could open up options for potentially curative treatment approaches that eliminate the antigen and thereby stop this ongoing reaction.

- Thank you for addressing this clinically relevant point. The revised Discussion makes a compelling case for antigen-targeted strategies. If feasible, consider briefly mentioning candidate antigens or classes of antigens (e.g., Borrelia-associated or autoantigenic targets) to increase translational relevance.

6. It is a common problem with rare diseases to work with the limited patient cohort (pcMZL n=9, pcFCL n=5, pcDLBCL-LT n=4, rB-LP n=5), the common issue which limits generalizability. In this particular study, the findings are not tested in an independent validation cohort and IHC validation (n=19) is not large enough for meaningful conclusions.

In order to strengthen our cohort, we were able to more than double the number of samples used for IHC validation (now n = 40). All of these new samples matched the findings from our scRNA-seq cohort.

Moreover, we were able to acquire two new samples from a patient with extensive rB-LP to show that the lesions are similar both with respect to clonal expansion and phenotypic composition.

Finally, we added a separate analysis of two patients where we acquired samples within 6 and 12 months respectively that also showed that the disease remained unchanged.

Thereby, we believe that we could arrive at a relevant sample size and further support the robustness of our methods.

- Thank you for significantly expanding the IHC validation cohort and including longitudinal and spatial comparisons. It significantly strengthened your manuscript. For transparency, you may wish to note the need for further validation in independent, multicenter cohorts as a future direction.

7. It is recommended to expand validation using publicly available datasets (Page 7, Lines 4-10): Validate scRNA-seq signatures against GEO datasets (e.g., GSE218861, EGAS00001006052, EGAS00001004904) and to perform bulk RNA-seq validation on an additional independent cohort. Additional technical validation (Page 10, Lines 3-6): the investigators can use spatial transcriptomics to confirm scRNA-seq findings at the tissue level. In addition, multiplex flow cytometry can validate protein expression of key markers in fresh patient samples.

As suggested we added an in-depth analysis and comparison to our data from the only available

scRNA-seq dataset on primary cutaneous B cell lymphoma (Rameyte *et al.*, new Figure 1H). These samples fully matched our own analysis.

Additionally, we were able to analyse a matched sample from one of our pcDLBCL-LT patients that showed a large number of presumably CD27+ B cells. This was fully confirmed by our IHC analysis further strengthening the accuracy of our scRNA-seq analysis (new Figure 2).

- Thank you for including the comparison to publicly available dataset (Rameyte *et al.*) These external data significantly reinforce the robustness of your conclusions and strengthens the manuscript.

8. Regarding statistical robustness (Page 14, Figure 4A), the investigators should consider applying Bonferroni or Benjamini-Hochberg corrections to prevent false positives in differential expression analysis and to use hierarchical mixed-effects models to account for patient-to-patient variability. Addressing these concerns would making findings more robust.

All p-values presented in the manuscript were corrected for multiple testing using the Benjamini-Hochberg ("FDR") correction. This is now clearly stated throughout the manuscript when the respective p-values are reported. Additionally, we adapted the methods section accordingly:

All p-values reported in the manuscript were corrected for multiple testing using the Benjamini-Hochberg ("FDR") correction

- Thank you for implementing multiple-testing corrections and clarifying this in the Methods and Results sections.

9. I would recommend improving figures and supplementary data integration including clarifying the labels and legends, improving references the supplementary figures in the main text, and overall improved data presentation (it is not always intuitive for a non-specialist). For example, I would recommend to improve clarity in key figures (Page 8, Figure 2A): add axis labels, clustering annotations, and improved color contrast; provide UMAP embeddings with more detailed explanations.

As suggested we re-created all supplementary figures and considerably extended the respective figure legends. We very much hope that these are now clearly tied to the main manuscript's text.

In the main manuscript, original figures 2 and 3 were removed and replaced with a new figure showcasing the analysis of our large IHC validation cohort (new Figure 2), as well as the in-depth comparison against systemic lymphoma (new Figure 3). Additionally, we added more graphical summaries to improve the presentation for non-specialists.

- Thank you for these substantial improvements. The updated figures and extended legends enhance clarity and accessibility, particularly for non-specialist readers.

10. Please, make sure that all supplementary figures are directly linked to main text (Page 6, Figure 1) by explicitly referencing which supplementary figures support key claims. Also, ensure figure legends provide enough information to be understandable on their own. It would be helpful to add summary diagrams (Page 12, Lines 18-22), such as a schematic of B-cell evolution in pcMZL vs. rB-LP would help non-specialists grasp findings more easily. Enhancing readability, accessibility, and figure integration, will make results easier to interpret.

As suggested we adapted all supplementary figures and their legends. Additionally, we added the suggested graphical summary to the new Figure 3.

- Thank you for adapting the supplementary figures and improving the legends. The B-cell evolution schematic is highly informative and will help a broader readership engage with your findings.

11. It is important to acknowledge study limitations and to justify cohort constraints. Currently, the study does not adequately acknowledge limitations of sample size, demographic bias, and antigen-driven hypotheses. For example, the antigen-driven persistence model is not directly tested, so that suggesting future multi-center studies to validate across diverse cohorts is warranted. To address antigen-driven persistence limitations (Page 15, Lines 8-12), functional validation studies to confirm antigen-driven B-cell expansion should be considered. Furthermore, peptide screening assays to identify putative antigens involved in pcMZL would be of great interest, though might be beyond the scope of this study. Also, longitudinal studies tracking B-cell evolution in pcMZL patients over time might be considered as future directions (Page 24, Conclusion).

As suggested, we added a direct comparison of two longitudinal samples from a patient with rB-LP and pcDLBCL-LT respectively (new Supplementary Figure 6). These showed that within recurring lesions within 6 and 12 months respectively the level of clonal expansion and phenotypic composition did not change.

Nevertheless, we also fully agree that our study can only be a starting point for larger studies that validate our findings. We believe that we provide the strongest evidence yet that indolent CBCL may be primarily antigen-driven reactions. Therefore, our data provides a good starting point for such in-depth and larger scale studies. We highlighted this in our discussion:

[...] Subsequent studies are needed to functionally validate this finding and identify these putatively driving antigens. [...]

- The revised manuscript provides promising evidence and outlines the need for further validation. To strengthen transparency, please consider explicitly acknowledging the limited sample size and demographic representation as part of the study's limitations.